# High-Calorie Diet Exacerbates the Crosstalk Between Gestational Diabetes and Youth-Onset Diabetes in Female Offspring Through Disrupted Estrogen Signaling

**DOI:** 10.3390/nu17132128

**Published:** 2025-06-26

**Authors:** Xinyu Jia, Xiangju Cao, Yuan Wang, Shuai Yang, Lixia Ji

**Affiliations:** 1Department of Pharmacology, School of Pharmacy, Qingdao University, Qingdao 266021, China; jiaxinyuzzz@163.com (X.J.); cxj19863348170@163.com (X.C.); wangyuanqdu@163.com (Y.W.); yangbb09252024@163.com (S.Y.); 2Key Laboratory of Maternal & Fetal Medicine of National Health Commission of China, Shandong Provincial Maternal and Child Health Care Hospital Affiliated to Qingdao University, Jinan 250014, China

**Keywords:** gestational diabetes mellitus, youth-onset type 2 diabetes, estrogen, estrogen receptors, high-calorie diet, insulin signaling pathway

## Abstract

**Background/Objectives**: Recent global trends highlight a concerning rise in youth-onset type 2 diabetes (YOT2D), with a marked female preponderance. We aim to explore the crosstalk between gestational diabetes mellitus (GDM) and YOT2D in female offspring. **Methods**: In vivo, GDM mice were induced by Western diet (WD), and their female offspring were fed normal diet or WD within 3 to 8 weeks. We continuously detected the glucose metabolism disorders, serum estradiol level (ELISA), and the process of ovarian maturation. Meanwhile, the dynamic changes in ERα and insulin signal in liver were monitored (qPCR, Western blot). In vitro, LO2 cells were treated with estradiol or ER antagonist BHPI to further explore the mechanism. **Results**: More than 85% of pregnant mice induced by WD were GDM models. The serum estradiol level in GDM offspring mice was decreased during sexual maturation, accompanied by marked oral glucose intolerance, insulin resistance, and even diabetes. The advance of sexual maturation and the decrease in serum estradiol in GDM offspring were mainly due to the downregulation of CYP19A1 in the ovaries, the reduced area of secondary follicles, and the increased number of atresia follicles, which could be greatly worsened by WD. Furthermore, GDM suppressed the protein levels of ERα, p-IRS-1, and p-Akt in liver tissue, that is, estrogen signals and insulin signaling were simultaneously weakened. WD further exacerbated the above changes. In vitro, estradiol upregulated the protein levels of ERα, p-IRS-1, and p-Akt in LO2 cells, while BHPI inhibited these changes. **Conclusions**: Maternal GDM promotes a high incidence of YOT2D in female offspring by affecting ovarian maturation, and a high-calorie diet exacerbates this process.

## 1. Introduction

Hyperglycemia in pregnancy mediates the intergenerational transmission of diabetes, and gestational diabetes mellitus (GDM) descendants are more likely to develop glucose metabolism disorders as they grow up [1]. According to the 10th edition of the Diabetes Atlas in 2021, more than about 21.1 million females (accounting for 16.80% of pregnant females) suffer from hyperglycemia, among which about 80.3% have GDM, that is, approximately one in six newborns are GDM descendants [2]. Common predisposing factors for a high incidence of GDM are obesity, advanced age, and a family history of diabetes. Youth-onset type 2 diabetes (YOT2D) occurs most frequently between the ages of 10 and 19 [3]. Adolescence is a critical period of sexual maturity, when the body is experiencing a dramatic rise in sex hormones and rapid maturation of various tissues and organs [4]. In 2021, an estimated 41,600 adolescents worldwide were newly diagnosed with type 2 diabetes (T2D) [5]. The global incidence rate (per 100,000 people) increased from 117.2 in 1990 to 183.4 in 2019 [6]. According to the current statistics, the incidence of YOT2D varies by ethnicity and region, with rates higher in Brazil, Mexico, the United States, and Canada but lower in Europe [2,7]. YOT2D has unique phenotype and pathological characteristics. Compared with adult T2D, YOT2D has a poorer glycemic trajectory, a higher failure rate on metformin monotherapy, and a faster decline in β cell function [8]. Patients with YOT2D are more likely to have cardiometabolic risk factors such as hypertension, elevated triglycerides, and central obesity compared with T1D of a similar age [9]. In addition, the incidence of YOT2D is higher in girls than in boys [10], indicating that large fluctuations in estrogen may be one of the causes of glucose metabolism disorders.

Estradiol (E2) is mainly synthesized in granulosa cells of ovarian follicles, and CYP19A1 catalyzes a critical step from testosterone to E2 [11]. A follicle has a clear trajectory in the development and maturation process, that is, a primitive follicle first develops into a primary follicle and then grows into a secondary follicle and a mature follicle [12]. However, the above process is easily affected by the microenvironment, and follicles in any state can develop into atretic follicles and gradually lose their activity [13]. The signaling mediated by Follicle-Stimulating Hormone Receptor (FSHR) promotes the regular growth of normal follicles and prevents the atresia of follicles. Additionally, FSHR may be involved in evaluating the functional status of atretic follicles [14]. As reported, follicle atresia was accompanied by upregulation of Caspase-3 and downregulation of Growth Differentiation Factor 9 (GDF9) and Anti-Müllerian Hormone (AMH) [13,15]. GDF9 is primarily expressed in oocytes, and promotes the growth and mitosis of oocytes and stimulates the proliferation of granulosa cells. Low level of GDF9 may disrupt the trajectory of follicle development, leading to follicle atresia [16]. AMH, mainly synthesized by granulosa cells, is a key indicator for evaluating ovarian reserve function. Hyperlipemia can damage the granulosa cells in the ovary and reduce the AMH secretion [17].

Clinical studies have confirmed a strong relationship between estrogen level and systemic insulin resistance. Premenopausal females have sensitive insulin responsiveness and low incidence of T2D compared with males of comparable age, but this advantage is interrupted after menopause, in part due to a decrease in circulating E2 [18]. Physiological concentrations of E2 are beneficial to insulin signaling, while E2 deficiency and/or resistance may lead to systemic insulin resistance. There are three main subtypes of estrogen receptors, estrogen receptor alpha (ERα), estrogen receptor beta (ERβ), and G protein-coupled estrogen receptor (GPER) [19]. Among them, ERα and ERβ located on the nuclear membrane mediate classical genetic effects, while GPER located on the cell membrane mediates rapid non-genetic effects [20]. The nuclear receptor ERα is the dominant subtype in hepatocytes closely related to metabolic syndrome. ERα regulates insulin signaling mainly through the ligand-dependent E2-ERα-PI3K-AKT pathway. The binding of E2 and ERα cascades the PI3K-Akt-FoxO1 signaling, which is a key point for sex differences in multiple metabolic diseases [21,22]. Recent studies have revealed that ERα can also promote the insulin sensitivity by inhibiting the ubiquitation-induced degradation of IRS-1 [23]. In hyper-metabolic stress, IRS-1 is usually ubiquitinated, recognized, and degraded by the proteasome, resulting in the obstruction of insulin signaling. By interacting with related proteins, ERα can interfere with the modification process of IRS-1 by ubiquitin enzymes, reduce the degradation of IRS-1, and maintain its stable expression in cells, thereby ensuring the normal transmission of insulin signaling and improving cells’ sensitivity to insulin [24].

Our previous study has confirmed that GDM offspring mice have lower ability to resist hypermetabolic stress in adulthood. Can GDM offspring cope with the complex and drastic hormone changes in adolescence normally? So far, there is a lack of systematic clinical data and basic research. In this study, we aim to investigate the correlation between maternal GDM and impaired glucose metabolism during sexual maturity in female-offspring mice and to explore its potential mechanisms from the perspective of estrogen disorders and insulin signaling.

## 2. Materials and Methods

### 2.1. Animal Experiment Design

Healthy 10-week-old female and male C57BL/6 mice (SCXK(Beijing)024-0003) were purchased from Beijing HuaFuKang Biotechnology Co., Ltd. (Beijing, China), and housed in the SPF Animal Experiment Center of Qingdao University (Qingdao, China). The ambient temperature was maintained at 22 ± 1 °C, and the relative humidity was maintained at 50 ± 1%. All animal experiments were conducted in accordance with protocols approved by the Animal Care and Use Committee of Qingdao University (approval number 20241009C578020250110194). The female mice were randomly divided into Group NC (*n* = 15) and Group GDM (*n* = 20). Mice in Group GDM were fed a Western diet (WD, 41% kcal from fat, 42.5% kcal from carbohydrate, 16.5% kcal from protein, H10141 HFK Ltd., Beijing, China) two weeks before mating until delivery, and those in Group NC were fed a normal diet (10% kcal from fat, 70% kcal from carbohydrate, 20% kcal from protein, H10010 HFK Ltd., Ramsgate, UK). All the male mice used for co-housing were healthy adult male mice, and they were provided with a normal diet. The co-housing process was carried out only at night, with a male-to-female ratio of 1:2. The vaginal smear was performed in the next morning. The simultaneous presence of sperm and keratinized epithelial cells was recorded as 0.5 in the embryo (E0.5). At 16.5 days of embryo (E16.5), an oral glucose tolerance test (OGTT) was performed on pregnant mice to identify the GDM model. At 3 weeks of age, the female-offspring mice were regrouped as follows: group NC-NC (*n* = 25), group normal-Western diet (NC-WD, *n* = 25), group GDM-NC (*n* = 25), and group GDM-WD (*n* = 25). From then on, female mice in group NC-NC and group GDM-NC were fed normal diet, and those in group NC-WD and group GDM-WD were fed with Western diet. From week 4 to week 8, vaginal smears were used to determine the estrous cycle, while blood, liver, and ovary samples were collected from each group during proestrus after the animals were anesthetized on each week (*n* = 5). For the serum estrodiol detection, 100 μL of blood was collected from the retro-orbital venous plexus in female mice in proestrus (*n* = 8–10). The OGTT and ITT experiments were conducted on female mice on week 5 (*n* = 8) and week 7 (*n* = 6) (Figure 1).

### 2.2. OGTT and Insulin Tolerance Test (ITT)

After 4 h of fasting, blood glucose at 0 min was measured using a Roche glucometer. Subsequently, the mice were gavaged with 20% glucose (2 g/kg), and blood glucose was measured at 30, 60, and 120 min. Glucose tolerance was evaluated by calculating the area under the OGTT curve (AUC). Subsequently, GDM models were identified based on OGTT results. Clinically, GDM is diagnosed based on OGTT results (75 g glucose) at 24–28 weeks of pregnancy. If the fasting blood glucose is ≥5.1 mmol/L, or the 1 h blood glucose is ≥10 mmol/L, or the 2 h blood glucose is ≥8.5 mmol/L, it can be diagnosed as GDM. After many previous experiments, we found that the blood glucose level in mice increased rapidly after glucose load (2 g/kg), almost reaching the highest value at 30 min, and dropped to near normal levels at 2 h. So diagnostic criteria for GDM mice model were fasting blood glucose ≥ 5.1 mmol/L, or 30 min blood glucose ≥ 10 mmol/L, or 60 min blood glucose ≥ 8.5 mmol/L.

After fasting for 4 h, the blood glucose at 0 min was measured first. Following subcutaneous injection of insulin solution (0.4 U/kg), blood glucose was measured at 40 and 90 min using a glucometer. The ITT blood glucose curves were plotted to calculate the percentage decrease in blood glucose at 40 min and assess the insulin sensitivity.

### 2.3. Immunohistochemistry (IHC)

Ovaries from offspring mice were fixed in 4% paraformaldehyde solution for 24–48 h and processed for paraffin embedding. The 4 μm tissue sections were dewaxed using the xylene-ethanol gradient method and then subjected to antigen retrieval by boiling in sodium citrate buffer. Endogenous peroxidase activity was quenched with hydrogen peroxide prior to blocking with goat serum. This was followed by overnight incubation at 4 °C with primary antibodies. The first antibodies were as follows: CYP19A1 (Cat# 16554-1-AP, 1:500, proteintech, Wuhan, China) and FSHR (Cat# 22665-1-AP, 1:500, proteintech, City, China). On the following day, sections were incubated with a secondary antibody at room temperature. The secondary antibody was as follows: immunohistochemical polymer secondary antibody (goat anti-rabbit), ready-to-use (Cat# G1302, Servicebio, Wuhan, China). DAB chromogen solution (Cat# G1212, Servicebio, Wuhan, China) was applied, and the reaction was terminated by washing with PBS. Sections were stained with hematoxylin (Cat# G1005, Servicebio, Wuhan, China), differentiated, blued, dehydrated, and mounted. Images were obtained using a panoramic tissue cell scanning analyzer (NIKON, Tokyo, Japan).

In order to evaluate the average staining intensity of CYP19A1 or FSHR in the ovaries, three ovary slices were selected from each group. Five images were collected from the top, down, left, right, and middle of the ovaries in the slices. Then we used the Pannoramic MIDI microscope and the integrated Slide Viewer software (2.5.0.143918) for image analysis to obtain the immunopositive areas of CYP19A1 or FSHR and the area of the ovaries in the images. The above procedure was performed by researchers Xiangju Cao and Yuan Wang, who were blinded to group assignment, with data subsequently analyzed by an independent researcher, Xinyu Jia.

The specific classification criteria for follicles at different types: In primary follicles, the follicular cells change from flat to cubic or columnar and proliferate from a single layer to multiple layers. A zona pellucida appears between the follicular cells and the oocyte. In secondary follicles, the number of follicular cell layers further increases, reaching 6 to 12 layers. Irregular cavities appear among the cells and gradually merge into a large cavity, which is called the follicular cavity. In atretic follicles, degenerative phenomena such as nuclear pyknosis and cytoplasmic lysis occur in the oocyte and follicular cells. The zona pellucida shrinks and thickens, and the follicular cavity collapses.

The specific operating procedures for immunohistochemical pictures were as follows: First, researchers Xiangju Cao and Yuan Wang classified follicles into morphological categories while blinded to group assignment and counted the follicles at each stage. The counting results were then independently subjected to statistical analysis by a separate researcher, Xinyu Jia.

### 2.4. Enzyme-Linked Immunosorbent Assay (ELISA)

Serum samples were collected from mice, and estradiol levels were quantified using a commercially available competitive ELISA kit (Kit: Cat# E-OSEL-M0008, Elabscience, Wuhan, China) according to the manufacturer’s instructions.

### 2.5. Real-Time Quantitative PCR (qPCR)

Total RNA from liver tissues was reverse-transcribed into cDNA by Hifair^®^III 1st Strand cDNA Synthesis SuperMix (YESEN, 11141ES60, Shanghai, China). Hieff^®^qPCR SYBR Green Master Mix (YESEN, 11202ES08, Shanghai, China) was used for qPCR reaction and analysis on an ABI real-time system (Thermo Fisher Scientific, Waltham, United States). The relative expression of the target gene was normalized to β-actin mRNA expression by the 2^−ΔΔCt^ method. The primers for ERα, ERβ, GPER, FSHR, Caspase-3, GDF, AMH, and β-actin were as follows:

m-ERα-F: CCCGCCTTCTACAGGTCTAAT;

m-ERα-R: CTTTCTCGTTACTGCTGGACAG,

m-ERβ-F: AACTCACCATCAAGCCTTACTTC;

m-ERβ-R: ACGGCTCTCTACATAGGAGGA,

m-GPER-F: ATGGATGCGACTACTCCAGC;

m-GPER-R: GGAAGAGGGCAATCACGTACT,

m-FSHR-F: CTCTGGGCCAGTCGTTTTAGAC;

m-FSHR-R: GCCTCAATGAGCATGACAAACTT,

m-Caspase-3-F: GAGGAGATGGCTTGCCAGAA;

m-Caspase-3-R: CTTGTGCGCGTACAGCTTCA,

m-GDF9-F: AATACCGTCCGGCTCTTCAGT;

m-GDF9-R: GGAAGAGGCAGAGTTGTTCAGAGT,

m-AMH-F: CACACAGAACCTCTGCCCTACTC;

m-AMH-R: AAGGCTTGCAGCTGATCGAT,

m-β-actin-F: CCTGTGGCATCCATGAAACTAC;

m-β-actin-R: CCAGGGCAGTAATCTCCTTCTG.

### 2.6. Western Blotting (WB)

Protease and phosphatase inhibitor cocktail was added to RIPA lysis buffer. Liver tissue proteins were extracted and quantified using the bicinchoninic acid (BCA) assay. Proteins were separated by SDS-PAGE (NCM Biological, P2012, Suzhou, China) and transferred to PVDF membranes (Merck Millipore, Billerica, MA, USA). Membranes were blocked with 5% skim milk and incubated overnight at 4 °C with the following primary antibodies: GAPDH (Cat# 60004-1-Ig, 1:10,000; Proteintech, Wuhan, China), ERα (Cat# ER1803-83, 1:1000; Huabio, Hangzhou, China), Akt (Cat# HA721870, 1:2000; Huabio, Hangzhou, China), p-Akt (Cat# F1644, 1:1000; Selleck, Houston, TX, USA), IRS-1 (Cat# PAB36510, 1:1000; Bioswamp, Shanghai, China), and p-IRS-1 (Cat# F1489, 1:1000; Selleck, Shanghai, China). The next day, the secondary antibodies were incubated at room temperature. The following secondary antibodies were used: Goat anti-mouse horseradish peroxidase (HRP) (Cat# Ab6789, 1:5000; Abcam, Cambridge, UK) and Goat anti-rabbit HRP (Cat# 111-035-003, 1:10,000; Jackson Immuno Research, West Grove, PA, USA). Protein signals were detected using an Ultra Sensitive enhanced chemiluminescence kit (GLPBIO GK10008, Beijing, China) and a chemical X-ray spectrometry imaging system (Bio-Rad, Hercules, CA, USA). Band intensities were quantified using Image Lab software (Bio-Rad 6.0) with GAPDH as a loading control.

### 2.7. Cell Culture and Treatment

Normal human hepatocytes (LO2, obtained from the Affiliated Hospital of Qingdao University) were cultured in DMEM (Procell, Wuhan, China) supplemented with 10% FBS (Lonza, Walkersville, MD, USA), penicillin (100 units/mL), and streptomycin (100 μg/mL) at 37 °C in an atmosphere containing 5% CO_2_.

LO2 cells were divided into a control group (0), a low-concentration E2 (Cat# GC11282, GLPBIO, Beijing, China) group (2 nM), a high-concentration E2 group (10 nM), and an ERα inhibitor group (BHPI, Cat# GC33011, GLPBIO, Montclair, CA, USA). Cells were sequentially treated with E2 for 24 h and BHPI for another 24 h. The protein levels of ERα, Akt, p-Akt, IRS1, and p-IRS1 were quantified by WB analysis.

### 2.8. Immunofluorescence (IF)

The LO2 cells were seeded into glass-bottomed dishes and fixed with 4% paraformaldehyde solution following induction. Cells were permeabilized with 0.1% Triton X-100, blocked with goat serum for 30 min at 37 °C, and incubated with primary antibody overnight at 4 °C. The used primary antibody was as follows: ERα (Cat# sc-8005, 1:1000, Santa, Santa Cruz, CA, USA). On the next day, the secondary antibody used for cell incubation was as follows: Alexa Fluor 488 anti-mouse (Cat# ab150113, 1:800, Abcam, Cambridge, UK). DAPI was incubated in the dark, and images were acquired using a high-speed super-resolution confocal microscope (Leica, Wetzlar, Germany).

### 2.9. Quantitative and Statistical Analysis

Statistical analysis was performed using GraphPad Prism 8.0. Results were expressed as mean ± standard deviation. Statistical differences were assessed using unpaired Student’s *t*-tests between two groups or two-way ANOVA between three or more groups. *p* < 0.05 was considered statistically significant.

## 3. Results

### 3.1. Maternal GDM Caused Glucose Metabolism Disorders in Female-Offspring Mice During Sexual Maturity

As shown in Figure 2A, the body weight of pregnant mice fed a Western diet was significantly higher than the control from E11 (*p* < 0.05). The OGTT results on E16.5 showed that compared with the control group, the blood glucose values at 0, 30, and 60 min and the area under the curve in GDM pregnant mice were significantly elevated (*p* < 0.05) (Figure 2B). Combined with clinical diagnostic criteria for GDM to screen GDM mouse models, 85% of pregnant mice fed WD met the requirements of GDM models. The 3-week-old female mice were regrouped for NC or Western diet induction. In the OGTT results at week 5, the blood glucose of GDM-WD mice increased markedly at 60 and 120 min after glucose load compared with the NC-NC levels (*p* < 0.01). In contrast, the blood glucose of GDM-NC mice showed a significant rise only at 60 min (*p* < 0.01) (Figure 2C). The area under the OGTT curve was markedly higher in both group GDM-NC and group GDM-WD, with the latter increasing more severely (*p* < 0.05). The ITT results at week 5 showed that compared with the NC-NC, the blood glucose of GDM-WD mice was significantly higher at 40 and 90 min after insulin injection (*p* < 0.01), and that of GDM-NC mice was obviously elevated only at 40 min (*p* < 0.01) (Figure 2D). Notably, the percentage of blood glucose drop at 40 min dropped markedly in groups GDM-NC and GDM-WD (*p* < 0.001), suggesting the development of insulin resistance in these female-offspring mice, which was more pronounced in GDM-WD offspring mice. Subsequently, OGTT and ITT (Figure 2E,F) were repeated again at week 7, which showed that the blood glucose levels of NC-WD, GDM-NC, and GDM-WD mice were significantly higher than the NC-NC levels at multiple time points in the OGTT and ITT results (*p* < 0.05). The same trend was also observed for the AUC in OGTT and the percentage of blood glucose drop at 40 min in ITT (*p* < 0.05). The GDM-WD mice exhibited the most severe degree of glucose intolerance and insulin resistance, which was comparable in group NC-WD and GDM-NC mice. During sexual maturation, the fasting blood glucose (FBG) in GDM-NC mice increased only slightly (*p* < 0.01), but the FBG in GDM-WD mice increased significantly since week 6, reaching as high as 7.84 mmol/L at week 8, which already met the diabetes criteria (Figure 2G). In addition, the body weight was calculated in weeks 3–8 (Figure 2H), which was similar in GDM-NC and NC-NC mice but increased markedly in GDM-WD and NC-WD mice during sexual maturity (*p* < 0.05). Taken together, although the body weight of GDM offspring mice on a normal diet remained almost normal during sexual maturity, they were still prone to obvious glucose intolerance and insulin resistance, indicating a pre-diabetic state. A high-fat diet could increase the body weight of GDM female-offspring mice, significantly disrupt glucose metabolism, and even develop fasting hyperglycemia, reaching the standard of diabetes.

### 3.2. Maternal GDM Caused Sexual Maturation Disruption and Ovarian CYP19A1 Downregulation in Female-Offspring Mice

Serum E2 levels were closely related to the estrous cycle. Proestrus was identified using the vaginal smear method and revealed a predominance of oval nucleated epithelial cells with few keratinized squamous cells (Figure 3A). The blood, ovaries, and livers of female-offspring mice were all collected in proestrus during sexual maturation. Figure 3B shows the dynamic changes in serum E2 of each group from weeks 4 to 8. In the NC-NC group, E2 gradually increased from week 4, peaked at about week 7, and then slightly declined and stabilized at week 8. During sexual maturation, the dynamic changes in serum estradiol in groups NC-WD and GDM-NC were similar to NC-NC, but the estradiol levels were lower than the control in the same week. However, in group GDM-WD, not only was the serum estradiol level lower than the control, but the peak was advanced to week 6. Subsequently, serum estradiol at week 7 of all groups was analyzed, and the results are shown in Figure 3C. Compared with the NC-NC level, serum estradiol levels in groups NC-WD, GDM-NC, and GDM-WD decreased by about 30.77%, 28.57%, and 34.29%, respectively.

CYP19A1, as a key enzyme in estrogen synthesis, was continuously detected in ovaries by immunohistochemistry from weeks 4 to 8 (Figure 3D). The expression of CYP19A1 in group NC-NC was progressively increasing from weeks 4 to 7 and slightly decreasing and stabilizing at week 8. The above changes were consistent with the dynamic changes in serum estradiol in NC-NC mice. At week 4, the protein levels of CYP19A1 in all groups were similar without significant differences. Starting from week 5, CYP19A1 protein in groups NC-WD, GDM-NC, and GDM-WD increased slowly, and as time went by, the gap between them and the NC-NC level gradually increased, with statistical differences (*p* < 0.05). Among them, group GDM-WD had the smallest increase in CYP19A1 protein, followed by groups GDM-NC and NC-WD (Figure 3E–I). Embryonic GDM exposure and high-calorie diet during growth are predisposing factors of ovarian dysfunction in female offspring. The combination of both further exacerbates the impact on ovarian maturation during sexual maturity.

### 3.3. Maternal GDM Caused Changes in the Area and Number of Follicle Types During Sexual Maturity

When analyzing the ovarian pathological sections, large differences in the sizes of follicles were found. Subsequently, we classified the follicles in the ovaries of the offspring mice and counted the area of each type of follicle. Figure 4A presents the areas of the primary follicle, secondary follicle, and atretic follicle of each group from weeks 4 to 8. In group NC-NC, the areas of three types of follicle progressively increased, and the secondary follicles increased fastest, with statistical significance since week 5 (*p* < 0.05). However, in group GDM-WD, the growth rate of secondary follicles decreased, and the fastest-growing type was the atresia follicle. Changes in group GDM-NC were similar to these described above but to a lesser extent. Compared within the same week, the areas of secondary follicles in groups NC-WD, GDM-NC, and GDM-WD were lower than the NC-NC level, and the areas in groups GDM-NC and GDM-WD decreased significantly from week 6 (*p* < 0.05). Immunohistochemistry results showed that most of the CYP19A1 enzyme was contained in granulosa cells of the secondary follicle, so a decline in the secondary follicular area also indicated the downregulation of CYP19A1 enzyme (Figure 4B). Meanwhile, we found that the area of atresia follicles and its increase rate in GDM-WD mice were faster than those in NC-NC mice, but there was no statistical difference. Atretic follicles usually occurred during the atresia of secondary follicles to maintain the normal development and homeostasis of follicles in the ovary. However, the hyperglycemic and low-grade inflammatory microenvironment could cause follicles at all levels to directly atrophy into atresia follicles, destroying the ovary homeostasis.

Figure 4C presented the number of primary follicles, secondary follicles, and atresia follicles during sexual maturation in female-offspring mice. As to the change pattern of primary follicles and secondary follicles, the number of both in group NC-NC gradually increased from week 4, reached the maximum number at week 7, and began to decrease at week 8. The change trend of the two types of follicles in groups NC-WD, GDM-NC, and GDM-WD was the same as that in group NC-NC, but the number of and increase in them were lower than the NC-NC level. Estradiol was mainly synthesized by granulosa cells in the secondary follicles, so the number of secondary follicles directly determined the serum estrogen level. The number of atresia follicles continued to increase from weeks 4 to 8 in all groups, but the increase was slow in group NC-NC and was the fastest in group GDM-WD. The number of atresia follicles in group GDM-WD was significantly higher than the NC-NC level since week 6 (*p* < 0.05), while that in groups GDM-NC and NC-WD increased markedly until week 8 (*p* < 0.05) (Figure 4D). Judging from the increase in atresia follicles, group GDM-WD was the most severe, followed by groups GDM-NC and NC-WD, suggesting that early exposure to GDM microenvironment had a greater impact on atresia follicles than high-fat diet during sexual maturity.

### 3.4. Maternal GDM Caused the Atresia of Follicles in Female-Offspring Mice During Sexual Maturity

In addition to being associated with CYP19A1, reduced serum estradiol levels in GDM female-offspring mice were associated with the number of atretic follicles. Staining for FSHR, a marker of granulocytes, as demonstrated in Figure 5A, the average staining intensity of FSHR in each group gradually increased within weeks 4–8, with the staining intensity in group NC-NC being the strongest and the other three groups being relatively weak (Figure 5B–F). Since week 6, the staining intensity of FSHR in groups GDM-WD, GDM-NC, and NC-WD were significantly lower than the NC-NC level, and that of group GDM-WD was still the lowest (*p* < 0.05). At the same time, qPCR was conducted to monitor the dynamic expression of FSHR within weeks 4–8, and similar results were obtained. As depicted in Figure 5G, compared with the NC-NC level, mRNA expression of FSHR in groups NC-WD, GDM-NC, and GDM-WD decreased significantly since week 6, with the lowest level in group GMD-WD (*p* < 0.05). In order to further clarify the mechanism of follicular atresia, the mRNA expression of follicular atresia-related genes including Caspase-3, GDF9, and AMH was measured at week 7. As to Caspase-3, mRNA levels in groups NC-NC, GDM-NC, and GDM-WD were significantly higher than the NC-NC level, with the most obvious increase in group GDM-WD (*p* < 0.05) (Figure 5H). For GDF9 and AMH, the trend was just the opposite. The mRNA expression of GDF9 and AMH was the highest in group NC-NC and significantly decreased in the other three groups, with the most obvious decrease in group GDM-WD (*p* < 0.05) (Figure 5I,J). The above results suggested that the increase in atretic follicles number caused by GDM or Western diet was also the cause of the decline of serum E2 levels.

### 3.5. Maternal GDM Impaired the Estrogen Receptor and Insulin Signaling in Liver Tissue

In this regard, three subtypes of estrogen receptor were detected by qPCR in the livers of NC-NC mice. As shown in Figure 6A, ERα was more abundantly expressed as the dominant estrogen receptor in hepatocytes. Afterward, Figure 6B presented the dynamic expression of ERα, and the varied trend was consistent with the change in serum E2. In group NC-NC, ERα mRNA gradually increased from week 4, peaked at about week 7, and then slightly decreased at week 8. The mRNA expression of ERα in group NC-WD, GDM-NC, and GDM-WD mice was lower than the NC-NC level at same week, and it peaked about one week ahead of the time in group GDM-WD.

The qPCR results of week 7 are shown in Figure 6C; ERα mRNA expression was significantly decreased in the other three groups compared with group NC-NC (*p* < 0.01), and the decline was the most severe in group GDM-WD. The differences in ERα expression between groups was further verified using the WB method. As shown in Figure 6D, the ERα protein level was the highest in group NC-NC, followed by groups NC-WD, GDM-NC, and GDM-WD (*p* < 0.05). GDM exposure combined with the Western diet greatly reduced the ER protein and mRNA expression in the liver. PI3K-Akt signaling was downstream of both the ERa signal and the insulin signal, and IRS is a downstream substrate of the insulin receptor. Subsequently, the phosphorylation levels of downstream Akt and IRS-1 were measured, and their change trends were basically consistent with that of ERα. WD and GDM could significantly decrease the phosphorylation of Akt and IRS-1, resulting in serious damage to estrogen signals and insulin signals.

### 3.6. The Interaction Between Estrogen and Insulin Signals In Vitro

To explore the interaction between estrogen receptor (ER) signaling and insulin signaling in hepatocytes, LO2 cells were treated with E2 and its inhibitor, BHPI. As depicted in Figure 7A, laser confocal microscopy revealed a notable increase in the fluorescence intensity of ERα in LO2 cells following E2 stimulation (*p* < 0.01) (Figure 7B). Our findings indicated that E2 promoted the protein level of ERα in LO2 cells in a concentration-dependent manner, and this upregulation could be effectively suppressed by BHPI.

Subsequently, we also detected that the phosphorylation levels of Akt and IRS-1 in LO2 cells. We observed that upon E2 intervention, the phosphorylation levels of both Akt and IRS-1 increased significantly. Conversely, BHPI significantly inhibited the elevated expression of phosphorylated Akt (p-Akt) and phosphorylated IRS-1 (p-IRS-1), suggesting an important role of E2 in modulating the insulin signaling pathway through ERα in hepatocytes (Figure 7C).

## 4. Discussion

In the clinic, adolescents (10–19 years old) are prone to obesity, hyperinsulinemia, insulin resistance, and other glucose metabolism disorders, and these metabolic disorders will gradually alleviate after puberty [25]. In fact, puberty is the period when physiological estrogen fluctuates the most, and it has a great impact on multiple organs/systems of the body [26]. The relationship between estrogen fluctuations and glucose metabolism disorders has long been reported, mainly focusing on menopause and pregnancy. Our previous research has confirmed that the embryonic GDM exposure is a high risk for a high incidence of metabolic diseases in the adult offspring [27]. Here, we wonder whether maternal GDM increases the degree of glucose metabolism disorders during sexual maturation in female offspring and whether this process is accompanied by abnormal fluctuations in serum estrogen levels.

The Western diet is closer to real life than a 60% high-fat diet, and mice are more receptive to the former, so we used the Western diet to induce more than 85% of pregnant mice to develop GDM. Female-offspring mice were regrouped and fed the normal diet or the Western diet at 3 weeks of age. The fasting blood glucose, OGTT, and ITT experiments were continuously performed during sexual maturity. We found that even if given a normal diet, GDM offspring mice developed oral glucose intolerance and insulin resistance since week 5, but the fasting blood glucose never increased throughout the sexual maturity process, so they were in a pre-diabetic state. However, the Western diet could aggravate the glucose metabolism disorders, and these GDM-WD offspring mice had significantly increased fasting blood glucose since week 6 and reached the diabetes criterion at week 8. The above results confirm the first conjecture that GDM female-offspring mice are more likely to develop glucose metabolism disorders during sexual maturity.

In order to assess the correlation between metabolic disorder and sex hormones, we continuously detected the dynamic changes in serum estradiol in the process of sexual maturation. It was found that the sexual maturation process of normal female mice occurred within 4–8 weeks of age, and the E2 peaked at week 7. Western diet or GDM exposure alone could reduce the serum E2 level during sexual maturation, but the time to peak remained unchanged. However, once the Western diet was combined with GDM, not only did the serum estradiol level in female offspring decrease, but also the estradiol peak was advanced to week 6. The simultaneous changes in systemic insulin resistance and low serum estradiol level during sexual maturation in GDM female offspring suggest some positive correlation between them.

How did the embryonic GDM exposure disrupt the sexual maturation process of female-offspring mice? Subsequently, the pathophysiological changes in the ovary were continuously monitored through the sexual maturity process. The ovaries of GDM female-offspring mice developed earlier during sexual maturation, but the CYP19A1 protein level in the ovaries was downregulated, which is a key enzyme that converts testosterone to estradiol. At the same time, statistical analyses were conducted on the area and number of various follicles in the ovary. We found that the secondary follicular area increased rapidly in NC-NC mice within weeks 4–8, while that in GDM-WD mice increased slowly and was significantly lower than the NC-NC level since week 6. However, the change in the number of atresia follicles was just the opposite; it increased slowly within weeks 4–8 in the NC-NC group but increased rapidly in the GDM-WD group. The development and maturation of follicles had clear regularity; the primitive follicle gradually developed into the primary follicle, the secondary follicle, and the mature follicle. During development, follicles of each type could directly transform into atresitic follicles and lose vitality. A significant increase in the number of atresitic follicles indicated that follicular development and maturation were impaired in the GDM female-offspring mice. Subsequent immunohistochemistry detected a decrease in the protein levels of FSHR located in granulosa cells, indicating that the biological function of granulosa cells was decreasing. Taken together, embryonic GDM exposure affected the development of multiple follicles during sexual maturation, resulting in a decrease in the number of granulosa cells, a decrease in the area of secondary follicles, and an increase in the number of atresia follicles. These pathological changes ultimately lead to an earlier sexual maturation process in GDM female-offspring mice but with a lower serum estradiol level. Previous studies [28] have reported that a high-fat diet in pregnancy caused obviously fewer oocytes in fetuses in the late embryo stage and increased primordial follicles in neonates with downregulation of AMH signaling. Combined with our research, it suggests that the impact of embryonic GDM exposure on the ovaries of female offspring begins early in life but is evident during sexual maturity.

Research on the correlation between estrogen levels and insulin resistance has long been conducted, especially during menopause [29]. Clinical data showed that as estrogen levels declined, the overall fat mass of menopausal females began to increase, and the sensitivity of peripheral tissues to insulin decreased, resulting in a rapid increase in the incidence of diabetes and cardiovascular disease [30]. In ovariectomized female rats, the phosphorylation levels of IRS-1 and Akt proteins were significantly reduced, systemic insulin resistance gradually became severe, and fasting hyperglycemia even occurred; these changes would be alleviated if E2 was supplemented in time [31]. The binding of E2 and ERα can directly activate the PI3K-Akt pathway [32,33], which is also a key downstream pathway for insulin receptor. To further explore the association between estrogen receptors and insulin signaling in puberty, liver tissue was collected as a target organ for follow-up research. We found that ERα was the dominant type of estrogen receptor in liver tissue, which was gradually upregulated during sexual maturity, peaked at week 7, and stabilized after 8 weeks. Early GDM exposure or high-fat diet in adolescence could inhibit the progressively high expression of ERα in liver tissue during sexual maturity, and the combined effects of both were more serious. Meanwhile, we found that insulin signaling in the livers of GDM female-offspring mice was suppressed, and the phosphorylation levels of key proteins IRS-1 and Akt were significantly reduced. If the GDM offspring mice were provided a Western diet, insulin signaling in the liver was further impaired.

Adolescents are in the period of the greatest fluctuations in sexual hormones in the lifetime. Youth onset type 2 diabetes is occurring at this stage, and its incidence is higher in girls than in boys [34]. Females are also prone to glucose metabolism disorders during pregnancy and menopause, and hormones fluctuate greatly during these two periods [35]. Violent fluctuations in estrogen are one of the stimulators of disorder in glucose metabolism [36]. In this study, we confirmed that eating a high-calorie diet during pregnancy increases the risk of gestational diabetes. Female-offspring mice with GDM are prone to glucose intolerance and insulin resistance during puberty even if they eat a normal diet, and a high-calorie diet will aggravate glucose metabolism disorders. Embryonic exposure to GDM affects the development and maturation of follicles in female-offspring mice during sexual maturity. The resulting low serum estrogen levels and downregulation of estrogen receptors are among the important reasons for the weakening of peripheral insulin signaling pathways. We hope these findings can offer a novel perspective for studying the youth-onset type 2 diabetes in GDM female offspring.

## 5. Conclusions

Embryonic GDM exposure affects the development and maturation of follicles during sexual maturation, causing the peak of serum estrogen to move forward, but serum estrogen level and ERα signaling are decreased. These are the reasons for the high incidence of glucose metabolism disorders in GDM female-offspring mice. However, judging from the root causes, the intrauterine GDM microenvironment is likely to have a serious impact on the development of the ovaries of female offspring during the embryonic stage, but this effect is obviously manifested during sexual maturity. Female offspring of GDM carry the mark of high metabolic stress from the embryo, and its adverse effects on ovarian development and maturation exacerbate the occurrence of youth-onset type 2 diabetes. In future research, we will focus on the impact of GDM’s hypermetabolic microenvironment on the development of the ovary of female offspring during the embryonic period and how to adjust the diet of pregnant females with GDM to minimize the adverse outcomes of GDM on the ovarian development of female-offspring mice.

## Figures and Tables

**Figure 1 nutrients-17-02128-f001:**
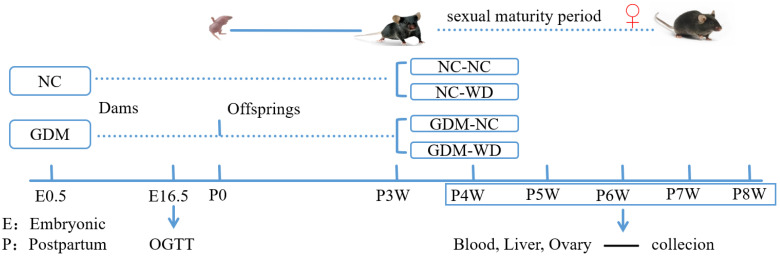
Flow chart of the overall animal experiment. First, the female mice were randomly divided into group NC (*n* = 15) and group GDM (*n* = 20). Mice in group NC were fed a normal diet, and those in group GDM were fed a Western diet. The OGTT test was conducted at 16.5 days of pregnancy, and the success rate of GDM modeling was about 85%. At 3 weeks of age, the female-offspring mice were regrouped as follows: group NC-NC (*n* = 25), group NC-WD (*n* = 25), group GDM-NC (*n* = 25), and group GDM-WD (*n* = 25). From then on, female mice in groups NC-NC and GDM-NC were fed the normal diet, and those in groups NC-WD and GDM-WD were fed with the Western diet. The estrous cycle was detected by vaginal smear every week from 4 to 8 weeks, and blood, ovary, and liver tissues were collected during proestrus (*n* = 5). The proestrus period was determined by the vaginal smear method. An amount of 100 μL of blood was collected from the posterior orbital venous plexus of 8–10 mice in each group, which was used for ELISA to determine the serum estrogen level. The OGTT and ITT experiments were conducted at week 5 (*n* = 8) and week 7 (*n* = 6).

**Figure 2 nutrients-17-02128-f002:**
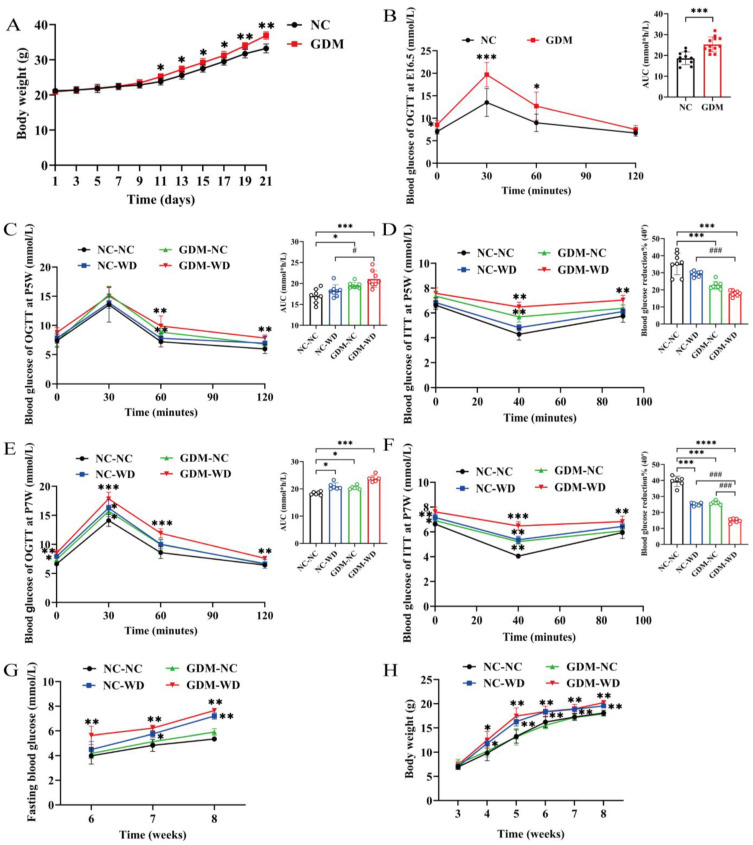
GDM caused glucose metabolism disorders in female-offspring mice during sexual maturity. (**A**) Gestational weight changes in C57 pregnant mice. (**B**) Plot of OGTT blood glucose and AUC at E16.5 in C57 pregnant mice. (**C**) Plot of OGTT blood glucose and AUC at week 5 in female-offspring mice. (**D**) ITT blood glucose and percentage of 40 min blood glucose drop at week 5 in female-offspring mice. (**E**) Plot of OGTT blood glucose and AUC at week 7 in female-offspring mice. (**F**) ITT blood glucose and percentage of 40 min blood glucose drop at week 7 in female-offspring mice. (**G**) Fasting blood glucose of female-offspring mice in weeks 6–8. (**H**) Body weight of female-offspring mice in weeks 3–8. The results are expressed as mean ± standard deviation, *n* = 9–12 (**A**,**B**), *n* = 6–8 (**C**–**F**), *n* = 4–8 (**G**), and *n* = 4–15 (**H**). * *p* < 0.05, ** *p* < 0.01, *** *p* < 0.001 vs. NC (**B**,**C**); * *p* < 0.05, ** *p* < 0.01, *** *p* < 0.001 **** *p* < 0.0001 vs. NC-NC, ^#^
*p* < 0.05, ^###^
*p* < 0.001 vs. GDM-WD (**D**–**H**).

**Figure 3 nutrients-17-02128-f003:**
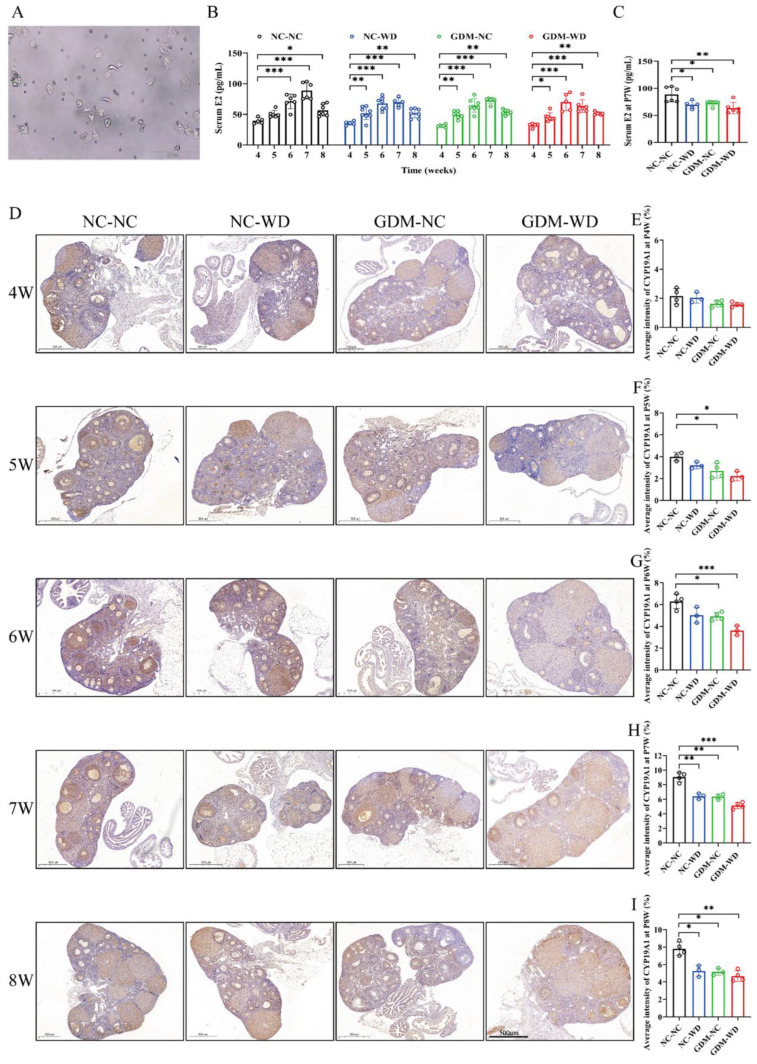
GDM disrupted sexual maturation and downregulated CYP19A1 enzyme in the ovaries of female-offspring mice. (**A**) Determination of proestrus by the vaginal smear method. (**B**) Serum E2 dynamics in female-offspring mice in weeks 4–8. (**C**) Serum E2 levels in female-offspring mice at week 7. (**D**) Representative images of CYP19A1 immunohistochemistry in ovaries of female-offspring mice in weeks 4–8. (**E**–**I**) Mean staining intensity of CYP19A1 within ovary tissues of female-offspring mice in weeks 4–8. The results are presented as mean ± standard deviation, *n* = 5–8 (**B**,**C**), and *n* = 3–4 (**D**–**I**). * *p* < 0.05, ** *p* < 0.01, *** *p* < 0.001 vs. 4 (**B**); * *p* < 0.05, ** *p* < 0.01, *** *p* < 0.001 vs. NC-NC (**C**–**I**); scale bar = 100 μm (**A**), scale bar = 500 μm (**D**).

**Figure 4 nutrients-17-02128-f004:**
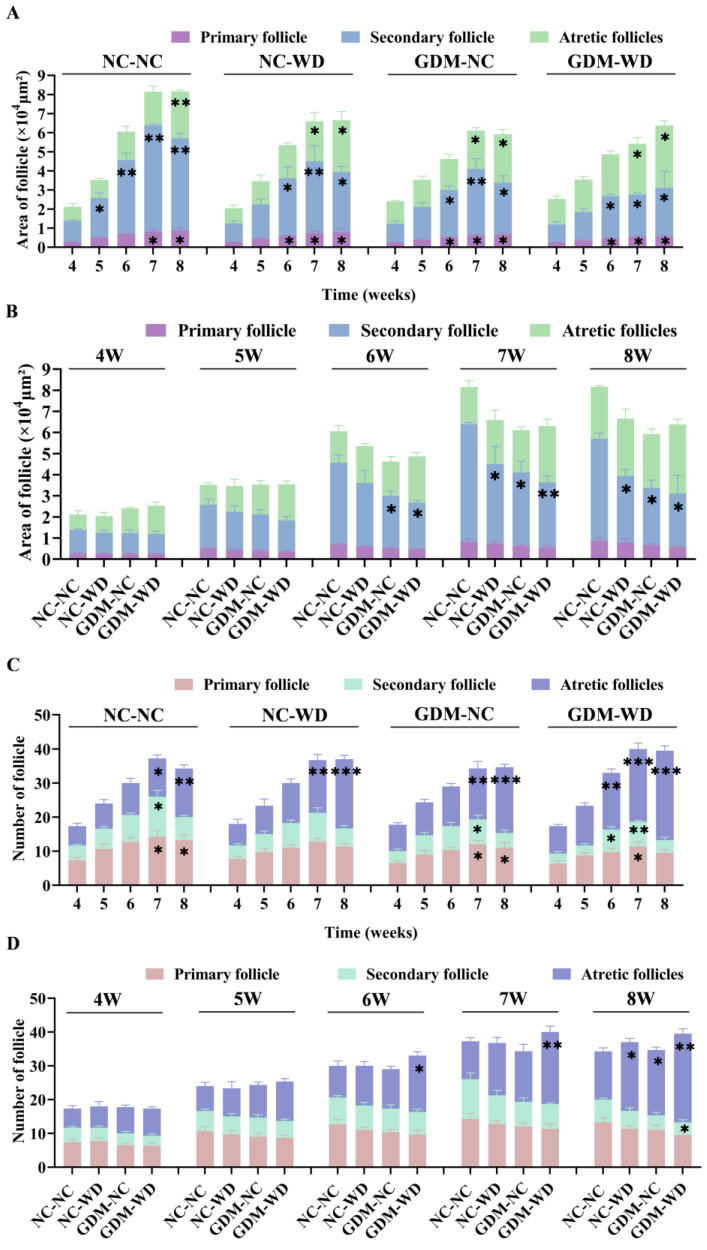
GDM resulted in changes in the area and number of follicle types during sexual maturity. (**A**) Dynamic changes in follicular area at all levels in the ovary in weeks 4–8. (**B**) Differences in follicular area at all levels between different groups. (**C**) Dynamic changes in number of follicles at all levels in the ovary in weeks 4–8. (**D**) Differences in the number of follicles at all levels between different groups. The results are presented as mean ± standard deviation, *n* = 3–4 * *p* < 0.05, ** *p* < 0.01, *** *p* < 0.001 vs. 4 (**A**,**C**); * *p* < 0.05, ** *p* < 0.01 vs. NC-NC (**B**,**D**).

**Figure 5 nutrients-17-02128-f005:**
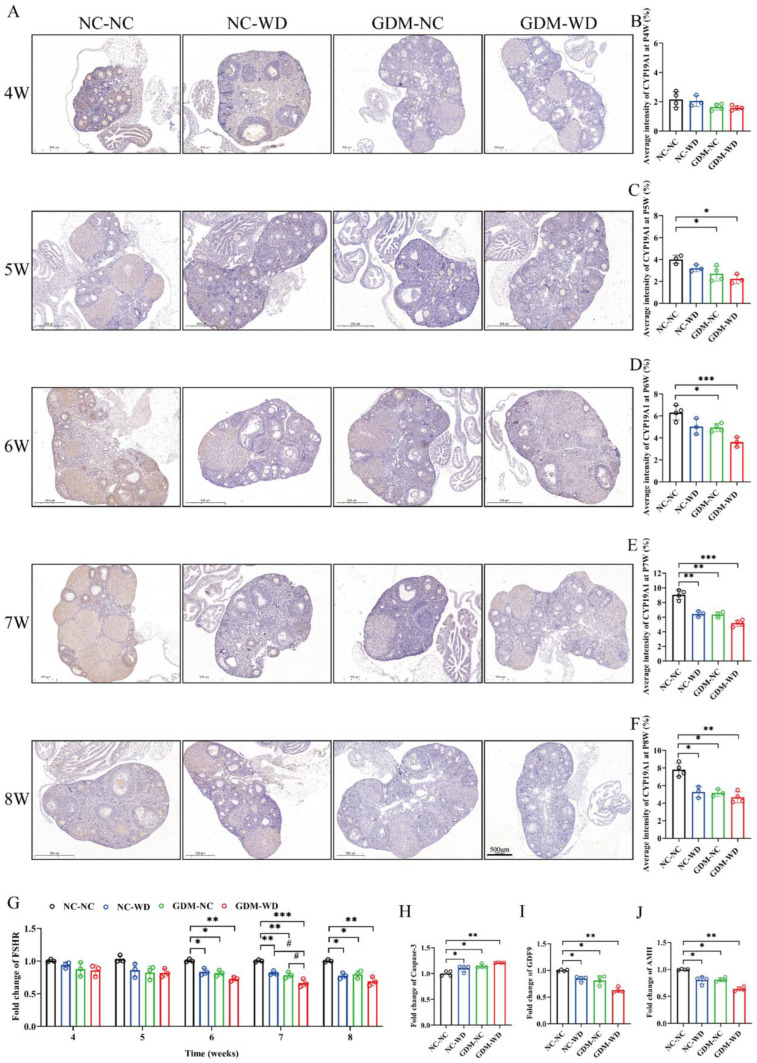
GDM led to an increase in the number of atretic follicles in female-offspring mice during sexual maturation. (**A**) Representative images of FSHR immunohistochemistry within the ovaries of female mice. (**B**–**F**) Average staining intensity of FSHR within the ovaries. (**G**) qPCR detection of FSHR mRNA expression in the ovary in weeks 4–8. (**H**) qPCR detection of Caspase-3 mRNA expression in the ovary at week 7. (**I**) qPCR detection of GDF9 mRNA expression in the ovary at week 7. (**J**) qPCR detection of AMH mRNA expression in the ovary at week 7. The results are presented as mean ± standard deviation, *n* = 3–4, * *p* < 0.05, ** *p* < 0.01, *** *p* < 0.001 vs. NC-NC, ^#^
*p* < 0.05 vs. GDM-WD, scale bar = 500 μm.

**Figure 6 nutrients-17-02128-f006:**
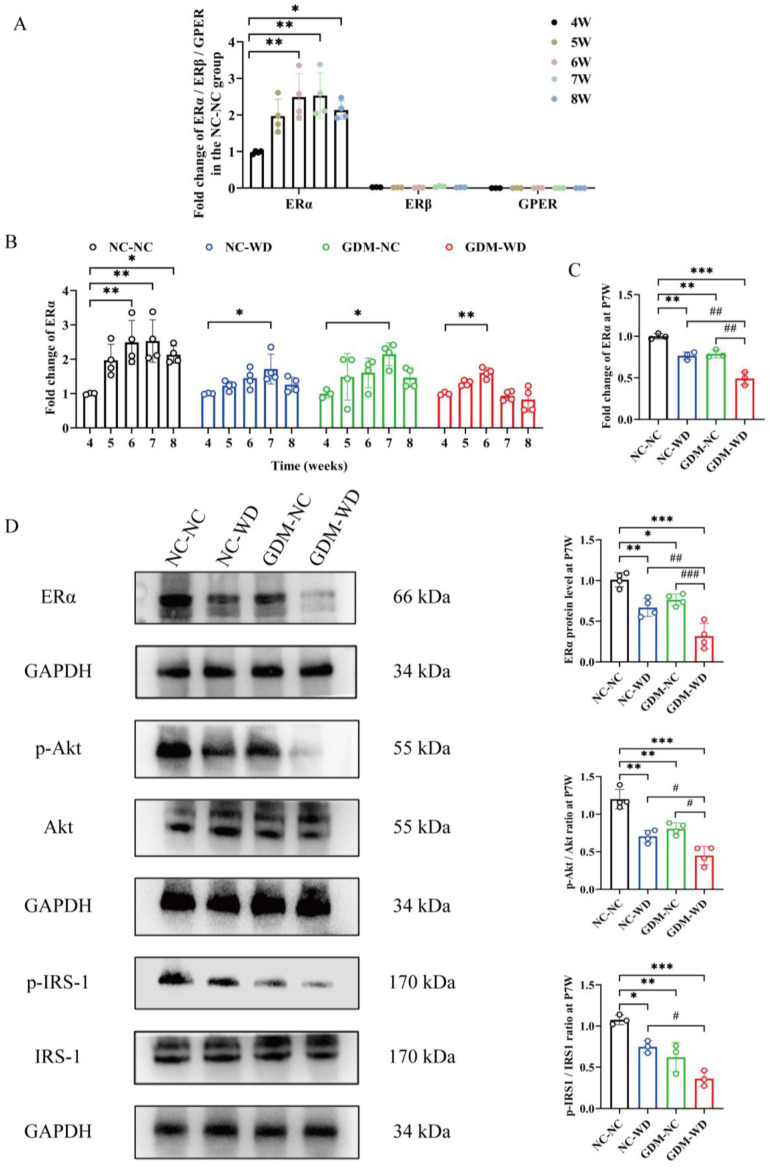
GDM impaired the estrogen receptor and insulin signaling in female-offspring mice during sexual maturation. (**A**) qPCR detection of estrogen receptor in the liver of group NC-NC. (**B**) qPCR detection of ERα mRNA expression in the liver. (**C**) qPCR detection of ERα mRNA expression in the liver at week 7. (**D**) WB analysis of ERα, p-Akt, Akt, p-IRS-1, and IRS-1 protein levels in the liver at week 7. The results are presented as mean ± standard deviation, *n* = 3–4, * *p* < 0.05, ** *p* < 0.01 vs. 4 (**A**,**B**); * *p* < 0.05, ** *p* < 0.01, *** *p* < 0.001 vs. NC-NC, ^#^
*p* < 0.05, ^##^
*p* < 0.01, ^###^
*p* < 0.001 vs. GDM-WD (**C**,**D**).

**Figure 7 nutrients-17-02128-f007:**
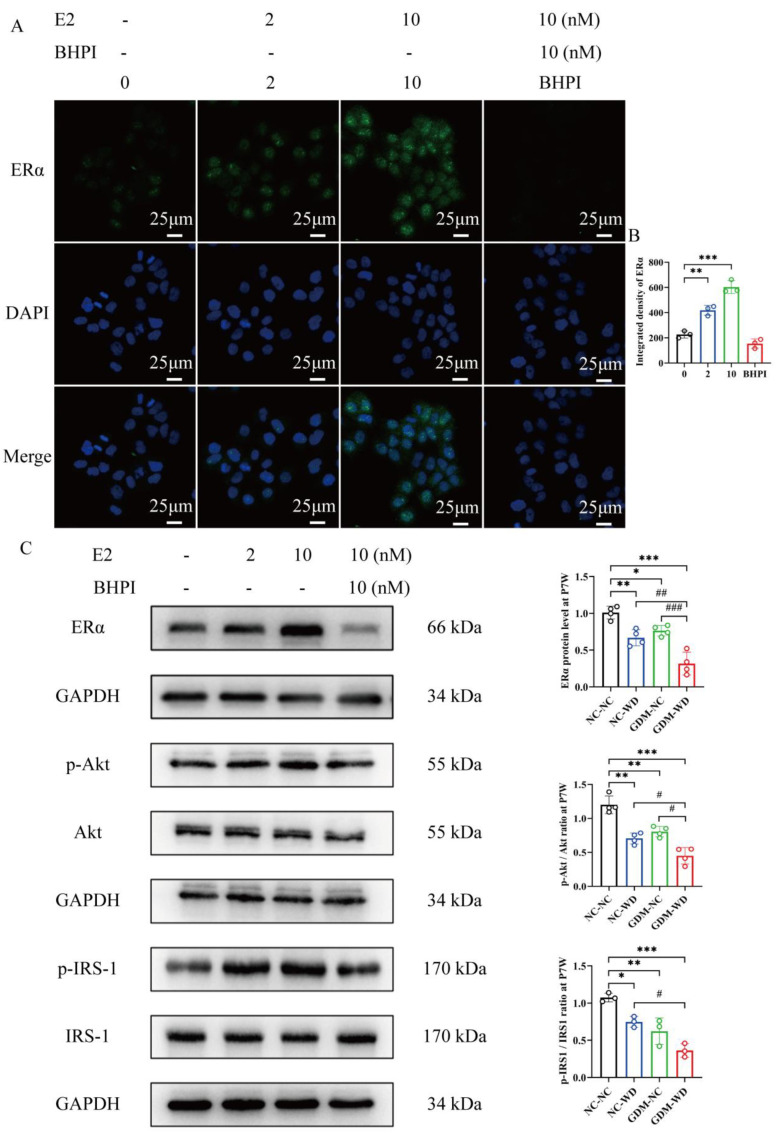
The correlation of the estrogen receptor and insulin pathway was validated in vitro. (**A**) Representative image of ERα immunofluorescence in LO2 cells. (**B**) Integrated density of ERα in LO2 cells. (**C**) WB analysis of ERα, p-Akt, Akt, p-IRS-1, and IRS-1 protein levels in LO2 cells. The results are presented as mean ± standard deviation, *n* = 3–4, * *p* < 0.05, ** *p* < 0.01, *** *p* < 0.001 vs. NC-NC, ^#^
*p* < 0.05, ^##^
*p* < 0.01, ^###^
*p* < 0.001 vs. GDM-WD.

## Data Availability

The original contributions presented in this study are included in the article. Further inquiries can be directed to the corresponding author.

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
