# Peer review of "High-Calorie Diet Exacerbates the Crosstalk Between Gestational Diabetes and Youth-Onset Diabetes in Female Offspring Through Disrupted Estrogen Signaling"

_nutrients, 2025, doi:10.3390/nu17132128_

Round 1
Reviewer 1 Report
Comments and Suggestions for Authors
This experimental study examines the impact of a Western diet on the induction of gestational diabetes mellitus (GDM) in pregnant mice, as well as its subsequent effects on the offspring. In the offspring, the authors evaluate multiple endpoints, including glucose metabolism, sexual maturation, ovarian histology, CYP19A1 regulation, and the hepatic expression of estrogen receptors and insulin signaling components. While the manuscript addresses a significant topic and presents substantial data, several issues must be resolved to enhance its clarity, depth, and scientific rigor before publication can be considered.
Comments on the Quality of English Language
- Although the manuscript offers valuable insights, several sections exhibit major grammatical issues that detract from its overall clarity. A comprehensive language revision is recommended to address these inconsistencies by refining sentence structures, enhancing word choice, and appropriately adjusting punctuation. These modifications are crucial for ensuring a smoother flow of information and improving the overall readability of the manuscript, thereby enabling readers to more effectively grasp the complex ideas being presented.
Comments and Suggestions
Abstract
- For consistency and clarity, the abstract should be revised so that the results are presented in the same format and detail as in the main results section of the manuscript. This alignment should include similar statistical details, descriptive terminology, and organizational structure to ensure that the abstract accurately and succinctly mirrors the findings detailed later in the paper.
Introduction
- Specific epidemiological data regarding both GDM and YOT2DM should be incorporated in the first paragraph of the introduction. Including prevalence rates, incidence figures, and other relevant metrics would quantitatively underscore the significance of these conditions, thereby providing readers with a clearer understanding of the extent and impact of the problem.
- The introduction should be revised to explicitly articulate the specific research gap that this experimental study aims to fill.
- Additionally, the study's objectives and research question need to be clearly delineated. Clearly articulating specific aims and expected contributions at the outset will provide a focused framework for readers and enhance the overall coherence of the narrative.
- In lines 46-51, the synthesis of E2 is well-established in literature and, thus, it should be eliminated to only the basic steps discussed later in the manuscript.
- The role of E2 in glucose metabolism should be elaborated in the introduction.
- In lines 52-54, the mention of "follicular atrophy" does not appear relevant to the content of the paragraph and, thus, it should be either removed or addressed in a separate paragraph where its relevance is more clearly justified.
- Given that ERα is a pivotal metabolic regulator, its role in promoting insulin sensitivity warrants further elaboration, reinforcing its significance as one of the main components of the research.
- The last paragraph of the introduction seems irrelevant to the introduction section, as it presents the methodology and results of the study.
- The introduction should briefly outline the diagnostic criteria employed in the methodology section and, specifically, regarding the OGTT and the ITT. It should also define the criteria for GDM and DM in general and indicate whether the established test values differ for the animal species used in this experiment.
- The introduction should also discuss the role of FSHR, Caspase-3, GDF9, AMH, BHPI, Akt and IRS-1 as signaling pathways regarding atretic follicles and insulin signaling, respectively.
Methods
- The selection of C57BL/6 mice as the animal species and the age of 10 weeks-old must be justified in the methods section.
- Please justify the selection of WD for the induction of GDM in pregnant mice.
- In lines 81-82 “all animals had free access to standard feed”, the “standard feed” is referring to the diet of choice for each study group? Please explain.
- A direct comparison of the “Western Diet” to the “normal diet” should be provided in the methods section.
- In line 92-94, a collection of “vaginal smears”, “blood, liver, and ovary samples” is described, but further elaboration on the exact timetable of sample collection and the total number of samples collected for each animal must be stated.
- Regarding the groups of female offspring, it is advisable to assign standardized labels (e.g., Group A, Group B, Group C, Group D, Control) to enhance clarity and readability throughout the manuscript.
- In line 101, what is “at the same time” referring to?
- In line 119, the manuscript appears to conflate hematoxylin/eosin (H&E) staining with immunohistochemistry. H&E staining is a conventional histological technique. Please clarify this distinction and ensure that readers accurately understand the methodologies employed.
- Figure 1A should be separated from the other results, as it illustrates the study’s methodology. Furthermore, it should be expanded to more effectively depict all aspects of the experimental procedures.
Results
- In line 193, the OGTT results are presented as a comparison between the Western diet and the normal diet. However, as outlined in the methods section (line 101), the OGTT was intended solely to validate the GDM mouse model rather than to serve as a direct comparison with the control group. Clarifying this distinction in the text will help maintain consistency between the methodology and the reported outcomes.
- In line 95, please clarify “most” when referring to the results of GDM induction strategy. It is recommended to specify the exact proportion or number of subjects in which the GDM induction was successful (e.g., "80% of the subjects" or "12 out of 15 mice").
- In lines 222-224, there is an incorrect interpretation of the results presented, as both GDM and control group when grouped with WD are presented with increased FBG. Please clarify accordingly.
- In line 253, please explain “continuously” referring to the exact timetable of the experimental procedures.
- When referring to "statistical differences" or "statistical significance" (e.g., in lines 260, 282, 292, 312), the corresponding p-values for each statistical test should be reported in the results section for every comparison made.
- In line 278, the manuscript describes a classification of the follicles without specifying the exact methodology. Please revise this section to clearly detail the criteria and procedures used for classifying the follicles.
- In line 281, the manuscript reports the area of follicles without specifying the methodology employed for this measurement. Please revise this section to clearly detail the criteria and procedures used for assessing the area of the follicles.
- In lines 288–290, please specify the method used to validate the CYP19A1 results (e.g., immunohistochemistry, ELISA, or Western Blotting).
- In line 294, “adverse microenvironment” must be further elaborated.
- In line 298, the term "judging" is used without specifying who evaluated the "change of pattern" (lines 298–299) or the criteria applied. Please provide further details regarding the evaluator and the assessment parameters in the methods section.
- Lines 303-305 are not results and, thus, the are irrelevant in this section.
- In line 326, please clarify who evaluated “the average staining intensity of FSHR” and specify the criteria used. Please provide further details regarding the evaluator and the assessment parameters in the methods section.
- In line 328-329, please clarify whether “protein levels” are referred to serum levels or intensity of staining.
- Sections 3.5 and 3.6 have the same title. Please revise.
Discussion
- In the first three paragraphs of the discussion section, there is extensive theoretical information that is not directly relevant to the discussion. Please revise these paragraphs to focus on the study’s findings and their implications.
- The discussion section is largely devoid of citations; please revise it to incorporate the necessary references.
- In order to further assess the discussion, the previous comments should be revised.
Conclusion
- There is no definitive conclusion in the manuscript. A clearly defined concluding section that summarizes the preclinical implications and outlines future directions would significantly enhance the manuscript.
General Comments
- The title of the manuscript appears misleading, as it suggests a focus on the “role of estrogen signaling in youth-onset diabetes in the offspring of gestational diabetes mellitus”. The study primarily examines the impact of a Western diet on the induction of GDM in pregnant mice and its subsequent effects on the offspring, including alterations in glucose metabolism, sexual maturation, ovarian histology, CYP19A1 regulation, and the hepatic expression of estrogen receptors and insulin signaling components.
Although the manuscript offers valuable insights, several sections exhibit major grammatical issues that detract from its overall clarity. A comprehensive language revision is recommended to address these inconsistencies by refining sentence structures, enhancing word choice, and appropriately adjusting punctuation. These modifications are crucial for ensuring a smoother flow of information and improving the overall readability of the manuscript, thereby enabling readers to more effectively grasp the complex ideas being presented.
Author Response
Reply to reviewer 1:
Comments 1: Although the manuscript offers valuable insights, several sections exhibit major grammatical issues that detract from its overall clarity. A comprehensive language revision is recommended to address these inconsistencies by refining sentence structures, enhancing word choice, and appropriately adjusting punctuation. These modifications are crucial for ensuring a smoother flow of information and improving the overall readability of the manuscript, thereby enabling readers to more effectively grasp the complex ideas being presented.
Response 1: Thank you so much for your time and efforts. We have corrected the grammar problems in the article and made it more fluent to read.
Comments 2: For consistency and clarity, the abstract should be revised so that the results are presented in the same format and detail as in the main results section of the manuscript. This alignment should include similar statistical details, descriptive terminology, and organizational structure to ensure that the abstract accurately and succinctly mirrors the findings detailed later in the paper.
Response 2: Thank you so much. We have referred to your opinions and revised the abstract. Modify as follows: Abstract: Background/Objectives: Recent global trends highlight a concerning rise in youth-onset type 2 diabetes (YOT2D), with a marked female preponderance. We aim to explore the crosstalk between gestational diabetes mellitus (GDM) and YOT2D in female offspring. Methods: In vivo, GDM mice were induced by Western diet (WD), and their female offspring were fed normal diet or WD within 3 to 8 weeks. We continuously detected the glucose metabolism disorders, serum estradiol level (ELISA) and the process of ovarian maturation. Meanwhile, the dynamic changes of ERα and insulin signal in liver were monitored (qPCR, Western blot). In vitro, LO2 cells were treated with estradiol or ER antagonist BHPI to further explore the mechanism. Results: More than 85% of pregnant mice induced by WD were GDM models. Serum estradiol level in GDM offspring mice was decreased during sexual maturation, accompanied by marked oral glucose intolerance, insulin resistance and even diabetes. The advance of sexual maturation and decrease of serum estradiol in GDM offspring were mainly due to down-regulation of CYP19A1 in ovaries, reduced area of secondary follicle, and increased number of atresia follicle, which could be greatly worsened by WD. Furthermore, GDM suppressed the protein levels of ERα, p-IRS-1 and p-Akt in liver tissue, that is, estrogen signals and insulin signaling were simultaneously weakened. WD further exacerbated the above changes. In vitro, estradiol up-regulated the protein level of ERα, p-IRS-1, and p-Akt in LO2 cells, while BHPI inhibited these changes. Conclusions: Maternal GDM promotes a high incidence of YOT2D in female offspring by affecting ovarian maturation, and a high-calorie diet exacerbates this process.
Comments 3: Specific epidemiological data regarding both GDM and YOT2DM should be incorporated in the first paragraph of the introduction. Including prevalence rates, incidence figures, and other relevant metrics would quantitatively underscore the significance of these conditions, thereby providing readers with a clearer understanding of the extent and impact of the problem.
Response 3: Thank you so much for your suggestion. We have emphasized the specific epidemiological data related to GDM and YOT2D in the introduction. This part is in lines 38-50.
Comments 4: The introduction should be revised to explicitly articulate the specific research gap that this experimental study aims to fill.
Response 4: The suggestions you put forward do have deficiencies for us. We have made revisions in the introduction part. The gap filled in this study is whether female offspring of GDM have a high incidence of glucose metabolism disorder diseases during sexual maturity and the role of estrogen signaling therein. Moreover, a high-calorie diet also has an impact on it, which is in lines 96-102.
Comments 5: Additionally, the study's objectives and research question need to be clearly delineated. Clearly articulating specific aims and expected contributions at the outset will provide a focused framework for readers and enhance the overall coherence of the narrative.
Response 5: Thank you so much for your suggestions. Our aim is to clarify the correlation between maternal gestational diabetes mellitus (GDM) and adolescent glucose metabolism disorders in female offspring mice.
Comments 6: In lines 46-51, the synthesis of E2 is well-established in literature and, thus, it should be eliminated to only the basic steps discussed later in the manuscript.
Response 6: We agree with the opinion you raised. The specific synthesis process of E2 has been deleted in the introduction, and only the last step of E2 synthesis has been mentioned (lines 58-59), because the CYP19A1 enzymes required for the last step are the contents that need to be studied in our manuscript.
Comments 7: The role of E2 in glucose metabolism should be elaborated in the introduction.
Response 7: Thank you so much for your suggestions. We have added the role of E2 in glucose metabolism in the introduction. This part is in lines 75-80.
Comments 8: In lines 52-54, the mention of "follicular atrophy" does not appear relevant to the content of the paragraph and, thus, it should be either removed or addressed in a separate paragraph where its relevance is more clearly justified.
Response 8: What you mentioned is so correct. The word "atrophy" is not quite appropriate to be used here. We have modified it to "atretic" and updated the content of this part in lines 59-63 of the text.
Comments 9: Given that ERα is a pivotal metabolic regulator, its role in promoting insulin sensitivity warrants further elaboration, reinforcing its significance as one of the main components of the research.
Response 9: Thank you for pointing out this issue. We have strengthened the content about the relationship between ERa and insulin sensitivity in the introduction, which is in lines 85-95.
Comments 10: The last paragraph of the introduction seems irrelevant to the introduction section, as it presents the methodology and results of the study.
Response 10: Thank you for your opinion. After careful consideration, we have revised and simplified the last paragraph of the introduction in lines 96-102.
Comments 11: The introduction should briefly outline the diagnostic criteria employed in the methodology section and, specifically, regarding the OGTT and the ITT. It should also define the criteria for GDM and DM in general and indicate whether the established test values differ for the animal species used in this experiment.
Response 11: Thank you for mentioning this point. We have made revisions in the manuscript. OGTT has judgment criteria, but ITT, which is used to detect insulin resistance, currently has no clear diagnostic criteria. OGTT diagnostic criteria: Clinically, GDM is diagnosed based on OGTT results (75 g glucose) at 24-28 weeks of pregnancy. If the fasting blood glucose is ≥ 5.1 mmol/L, or the 1-hour blood glucose is ≥ 10 mmol/L, or the 2-hour blood glucose is ≥ 8.5 mmol/L, it can be diagnosed as GDM. After many previous experiments, we found that blood glucose level in mice increased rapidly after glucose load (2 g/kg), almost reaching the highest value at 30 minutes, and dropped to near normal levels at 2 hour. So diagnostic criteria for GDM mice model is fasting blood glucose ≥ 5.1 mmol/L, or 30 min blood glucose ≥ 10 mmol/L, or 60 min blood glucose is ≥ 8.5 mmol/L. This part of the content has been added to lines 136-143 in the text.
Comments 12: The introduction should also discuss the role of FSHR, Caspase-3, GDF9, AMH, BHPI, Akt and IRS-1 as signaling pathways regarding atretic follicles and insulin signaling, respectively.
Response 12: Thank you for your valuable suggestions. We have made revisions in the introduction. FSHR, Caspase-3, GDF9 and AMH are mainly regarded as genes related to follicular atresia in our manuscript, so we have supplemented them in lines 64-74. BHPI is only used as an inhibitor of estrogen receptor α (ERα). Akt and IRS-1 are proteins related to the insulin signaling pathway, and we supplemented them in lines 85-95.
Comments 13: The selection of C57BL/6 mice as the animal species and the age of 10 weeks-old must be justified in the methods section.
Response 13: Thank you so much for your suggestion. The choice of animal species is very important to the credibility of experimental results. C57BL/6 mice are a recognized mouse species that is often used for research on metabolic diseases due to its sensitivity to a high-calorie diet. The mice had become adults at 10 weeks of age. After entering the Animal Center of Qingdao University, the mice first adapted to the environment for one week, and were divided into groups-NC and GDM groups at 11 weeks of age. Since female mice in the GDM group needed to undergo Western diet induction 2 weeks in advance before being cooped up, the female mice were at 13 weeks old at this time, which was the optimal age for reproduction.
Comments 14: Please justify the selection of WD for the induction of GDM in pregnant mice.
Response 14: Thanks. High-fat diet (HFD) feeding is a recognized method of inducing diabetes. Common high-fat diet types are 60%-HFD, 45%-HFD and Western diet. These three diets can cause mice to develop significant obesity, oral glucose intolerance, insulin resistance and later elevated fasting blood glucose. We have used 60%- and 45%-HFD to induce GDM models in mice and rats. Although the GDM model was successful, the animals were in poor condition after 2-3 weeks of feeding and were resistant to HFD, especially the 60% HFD with the highest fat content. We later compared the 60% HFD, 45% HFD and WD formulas (table1) and found that WD had the lowest fat content at 41%, the highest carbohydrate content at 42.5%, and also increased bile acid by 0.15%. The WD formula is closer to a high-calorie diet in real life. After multiple experiments and comparisons, it was found that the compliance of mice to WD feed was significantly higher than that of 60% HFD, and the status of animals fed WD in the third trimester of pregnancy was relatively better, and the incidence of stillbirth and dystocia during delivery was also reduced.
Table 1 Comparison of three types of feed
|
products |
Standard feed (H10010) |
Western diet (H10141) |
60% high-fat (H10060) |
||||||||||
|
mass ratio g % |
energy ratio kcal % |
mass ratio g % |
energy ratio kcal % |
mass ratio g % |
energy ratio kcal % |
||||||||
|
protein |
19.2 |
20 |
20 |
16.5 |
26 |
20 |
|
||||||
|
carbohydrate |
67.3 |
70 |
50 |
42.5 |
26 |
20 |
|
||||||
|
fat |
4.3 |
10 |
21 |
41 |
35 |
60 |
|
||||||
|
cholesterol |
0 |
0 |
1.5 |
0 |
0 |
0 |
|
||||||
|
kcal/g |
3.85 |
|
4.7 |
|
5.24 |
|
|
||||||
Other literature on WD-induced diabetes is as follows:
[1] Herzl E, Schmitt E E, Shearrer G, et al. The Effects of a Western Diet vs. a High-Fiber Unprocessed Diet on Health Outcomes in Mice Offspring[J]. Nutrients, 2023, 15(13).
[2] Elshareif N, Gornick E, Gavini C K, et al. Comparison of western diet-induced obesity and streptozotocin mouse models: insights into energy balance, somatosensory dysfunction, and cardiac autonomic neuropathy[J]. Frontiers in Physiology, 2023, 14.
[3] Lipovšek S, Dolenšek J, Dariš B, et al. Western diet-induced ultrastructural changes in mouse pancreatic acinar cells[J]. Frontiers in Cell and Developmental Biology, 2024, 12.
Comments 15: In lines 81-82 “all animals had free access to standard feed”, the “standard feed” is referring to the diet of choice for each study group? Please explain.
Response 15: Thank you so much for your review. “standard feed” means feeding normal diet. The specific feed conditions of the NC and GDM groups have been written in detail here. The female mice were randomly divided into Group NC (n=15) and Group GDM (n=20). Mice in Group GDM were fed a Western diet (41% kcal from fat, 42.5% kcal from carbohydrate, 16.5% kcal from protein, H10141 HFK Ltd) two weeks before mating until delivery, and those in Group NC were fed a normal diet (10% kcal from fat, 70% kcal from carbohydrate, 20% kcal from protein, H10010 HFK Ltd.). Lines 111-115.
Comments 16: A direct comparison of the “Western Diet” to the “normal diet” should be provided in the methods section.
Response 16: Thank you so much for pointing out the deficiencies in my method part. We have made revisions. We have listed the specific ingredients and contents of "Western diet" and "normal diet" in the method. Western diet (WD; 41% kcal from fat, 42.5% kcal from carbohydrate, 16.5% kcal from protein, H10141 HFK Ltd), while Group NC mice were fed a normal diet (NCD; 10% kcal from fat, 70% kcal from carbohydrate, 20% kcal from protein, H10010 HFK Ltd.). Lines 112-115.
Comments 17: In line 92-94, a collection of “vaginal smears”, “blood, liver, and ovary samples” is described, but further elaboration on the exact timetable of sample collection and the total number of samples collected for each animal must be stated.
Response 17: Thank you for conducting an in-depth review of our method. We have added the modified content in the method section. From week 4 to week 8, vaginal smears were used to determine the estrous cycle, while blood, liver, and ovary samples were collected from each group during proestrus on each week (n = 5). For the serum estrodiol detection, 100 μL of blood was collected from the retro-orbital venous plexus in female mice in proestrus (n = 8 - 10); The OGTT and ITT experiments were conducted on female mice on week 5 (n = 8) and week 7 (n = 6) .Lines 125-130.
Comments 18: Regarding the groups of female offspring, it is advisable to assign standardized labels (e.g., Group A, Group B, Group C, Group D, Control) to enhance clarity and readability throughout the manuscript.
Response 18: Thank you for your comments. The group name has been revised in the full text based on your suggestions, as follows. At 3 weeks of age, the female-offspring mice were regrouped as follows: Group Normal-Normal (NC-NC, n = 25), Group Normal-Western diet (NC-WD, n = 25), Group GDM-Normal (GDM-NC, n = 25), and Group GDM-Western diet (GDM-WD, n = 25). Lines 123-125. In addition, other places related to grouping in the full text have also been modified and marked in the text.
Comments 19: In line 101, what is “at the same time” referring to?
Response 19: Thank you for the problem you found. The use of "at the same time" here is indeed a bit inappropriate and has been changed to "subsequently". Line 135.
Comments 20: In line 119, the manuscript appears to conflate hematoxylin/eosin (H&E) staining with immunohistochemistry. H&E staining is a conventional histological technique. Please clarify this distinction and ensure that readers accurately understand the methodologies employed.
Response 20: Thank you for your suggestions. In this study, immunohistochemical methods were used to detect the expression of CYP19A1 and FSHR in ovary. We first embedded tissue blocks and sectioned them according to conventional methods, with a thickness of 4 μm, then incubated the primary and secondary antibodies successively, and finally stained the cell nuclei with hematoxylin. This method is the routine process of immunohistochemistry, except that some of the steps are the same as HE staining.
Comments 21: Figure 1A should be separated from the other results, as it illustrates the study’s methodology. Furthermore, it should be expanded to more effectively depict all aspects of the experimental procedures.this distinction and ensure that readers accurately understand the methodologies employed.
Response 21: Thank you for your suggestion, which will be very helpful to readers in getting the overall idea of animal experiments. Figure 1A has been taken out separately as Figure 1, and has been expanded to describe in detail the specific process of the experimental procedures. Lines 288-301
Comments 22: In line 193, the OGTT results are presented as a comparison between the Western diet and the normal diet. However, as outlined in the methods section (line 101), the OGTT was intended solely to validate the GDM mouse model rather than to serve as a direct comparison with the control group. Clarifying this distinction in the text will help maintain consistency between the methodology and the reported outcomes.
Response 22: Thank you for your comments. This part of the OGTT content has been revised. OGTT is usually used for early evaluation of oral glucose tolerance, which is mainly evaluated by measuring blood sugar levels at multiple time points before and after glucose load. Clinically, GDM is also diagnosed based on OGTT results (75g glucose) at 24-28 weeks of pregnancy. If the fasting blood glucose is ≥ 5.1 mmol/L, the 1-hour blood glucose is ≥ 10 mmol/L, or the 2-hour blood glucose is ≥ 8.5 mmol/L, it can be diagnosed as GDM. After many previous experiments, we found that blood glucose level in mice increased rapidly after glucose load (2 g/kg), almost reaching the highest value at 30 minutes, and dropped to near normal levels at 2 hour. So diagnostic criteria for GDM mice model is fasting blood glucose ≥ 5.1 mmol/L, or 30min blood glucose ≥ 10 mmol/L, or 60 min blood glucose is ≥ 8.5 mmol/L. Therefore, in this part of the result analysis, OGTT results reflects two aspects: one is whether each pregnant mouse meets the GDM standard, and the other is the overall difference in glucose metabolism disorders between the NC group and the GDM group. Lines 136-143.
Comments 23: In line 95, please clarify “most” when referring to the results of GDM induction strategy. It is recommended to specify the exact proportion or number of subjects in which the GDM induction was successful (e.g., "80% of the subjects" or "12 out of 15 mice").
Response 23: Thanks to the problem you found, "most" is indeed very inaccurate. According to the method adopted above, based on the OGTT results tested on E16.5 days, the success rate of the GDM model is about 85%, and "most" has been changed to 85% in the manuscript. Lines 257-258.
Comments 24: In lines 222-224, there is an incorrect interpretation of the results presented, as both GDM and control group when grouped with WD are presented with increased FBG. Please clarify accordingly.
Response 24: Thank you for your question, there are no mistakes here. Female offspring mice in both NC-WD and GDM-WD groups showed significant weight gain and fasting blood glucose increase after 5 weeks of WD feeding, but the increase was more significant in the GDM-WD group (Figure. 2G, 2H). Because WD is a high-calorie diet, normal control mice will also develop typical manifestations of diabetes after the induction weeks of WD diet. However, during the WD induction period from week 3 to 8, the degree of glucose metabolism disorder in GDM offspring mice was significantly higher than that in normal offspring mice.
Comments 25: In line 253, please explain “continuously” referring to the exact timetable of the experimental procedures.
Response 25: Thank you for your question. “continuously”means from week 4 to week 8. This time has already been reflected in the manuscript. Line 331.
Comments 26: When referring to "statistical differences" or "statistical significance" (e.g., in lines 260, 282, 292, 312), the corresponding p-values for each statistical test should be reported in the results section for every comparison made.
Response 26: Thanks to the problem you found, P values for multiple inter-group comparisons have been marked in the manuscript.
Comments 27: In line 278, the manuscript describes a classification of the follicles without specifying the exact methodology. Please revise this section to clearly detail the criteria and procedures used for classifying the follicles.
Response 27: Thanks to the problem you found, This part is indeed lacking and has been described in detail in the method (lines 170-182). The specific classification criteria for follicles at different levels: In primary follicles, the follicular cells change from flat to cubic or columnar, and proliferate from a single layer to multiple layers. A zona pellucida appears between the follicular cells and the oocyte. In secondary follicles, the number of follicular cell layers further increases, reaching 6 to 12 layers. Irregular cavities appear among the cells and gradually merge into a large cavity, which is called the follicular cavity. In atretic follicles, degenerative phenomena such as nuclear pyknosis and cytoplasmic lysis occur in the oocyte and follicular cells. The zona pellucida shrinks and thickens, and the follicular cavity collapses. Specific operating procedures for immunohistochemical pictures: First, two investigators classified the follicles and counted the follicles at all levels without knowing the grouping. The counting results were processed statistically by the third researcher.
Comments 28: In line 281, the manuscript reports the area of follicles without specifying the methodology employed for this measurement. Please revise this section to clearly detail the criteria and procedures used for assessing the area of the follicles.
Response 28: Thanks. The method of follicular area is similar to that of follicle amount. First, two investigators classified the follicles and measured the area of follicles at all levels without knowing the grouping. The area results were processed statistically by the third researcher. Lines 170-182.
Comments 29: In lines 288–290, please specify the method used to validate the CYP19A1 results (e.g., immunohistochemistry, ELISA, or Western Blotting).
Response 29: Thanks. In this study, CYP19A1 were detected by Immunohistochemistry, which was specified in this part of the manuscript. Lines 148-169 and 331.
Comments 30: In line 294, “adverse microenvironment” must be further elaborated.
Response 30: Thanks to the problem you found. GDM provides poor uterine microenvironment for the offsprings, which are characterized by hyperglycemia and low-grade metabolic inflammation. We have changed “adverse microenvironment” to “the hyperglycemic and low-grade inflammatory microenvironment”.Line 372.
Comments 31: In line 298, the term "judging" is used without specifying who evaluated the "change of pattern" (lines 298–299) or the criteria applied. Please provide further details regarding the evaluator and the assessment parameters in the methods section.
Response 31: Thanks. The result of this description is still the comparison of follicular area at all levels, and the evaluation criteria have been listed in the method. The word “judging” is not very accurate and has been changed to “As to”. Line 375.
Comments 32: Lines 303-305 are not results and, thus, the are irrelevant in this section.
Response 32: Thanks. The previous sentence only describes the change trend of primary follicles and secondary follicles in Group NC-NC, followed by the description of three other groups (similar to NC-NC), and if this sentence is deleted, it will lack the results of the changes in the three other groups. Lines 377-380.
Comments 33: In line 326, please clarify who evaluated “the average staining intensity of FSHR” and specify the criteria used. Please provide further details regarding the evaluator and the assessment parameters in the methods section.
Response 33: In order to evaluate the average staining intensity of FSHR or CYP19A1 in the ovaries, three ovary slices were selected from each group. Five images were collected from the top, down, left, right and middle of the ovaries in the slices. Then we used the Pannoramic MIDI microscope and the integrated Slide Viewer software for image analysis to obtain the immunopositive areas of CYP19A1 or FSHR and the area of the ovaries in the images. The above operation was operated by two researchers without knowing the grouping.. Then the third researcher analyzed the data. Detailed operation steps have been added to the method. Lines 163-169.
Comments 34: In line 328-329, please clarify whether “protein levels” are referred to serum levels or intensity of staining.
Response 34: Thanks. The “protein levels” here is the intensity of staining detected by immunohistology, which is clearly stated in this article. Line 405.
Comments 35: Sections 3.5 and 3.6 have the same title. Please revise.
Response 35: Thank you for the mistake you found. The title of 3.6 has been modified. Modify to :The interaction between estrogen and insulin signals in vitro. Line 460.
Comments 36: In the first three paragraphs of the discussion section, there is extensive theoretical information that is not directly relevant to the discussion. Please revise these paragraphs to focus on the study’s findings and their implications.
Response 36: Thank you so much for your suggestions. We have revised the first paragraph of the discussion. Lines 480-490 in the text.
Comments 37: The discussion section is largely devoid of citations; please revise it to incorporate the necessary references.
Response 37: Thanks to the valuable comments from the reviewers, we have updated the references.
Comments 38: In order to further assess the discussion, the previous comments should be revised.
Response 38: Thank you so much for reviewing my article in your busy schedule. We have made revisions to the discussion section.
Comments 39: There is no definitive conclusion in the manuscript. A clearly defined concluding section that summarizes the preclinical implications and outlines future directions would significantly enhance the manuscript.
Response 39: Thank you so much for your suggestion. We have added the relevant content in lines 576-578.
Comments 40: The title of the manuscript appears misleading, as it suggests a focus on the “role of estrogen signaling in youth-onset diabetes in the offspring of gestational diabetes mellitus”. The study primarily examines the impact of a Western diet on the induction of GDM in pregnant mice and its subsequent effects on the offspring, including alterations in glucose metabolism, sexual maturation, ovarian histology, CYP19A1 regulation, and the hepatic expression of estrogen receptors and insulin signaling components.
Response 40: Thank you. We have modified the title of the article. The title has been modified to: High-calorie Diet Exacerbates the Crosstalk Between Gestational Diabetes and Youth-onset Diabetes in Female Offspring through Disrupted Estrogen Signaling.
Reviewer 2 Report
Comments and Suggestions for Authors
In this study, the authors investigated the role of estrogen signaling in youth-onset diabetes in the off-springs of dams with gestational diabetes (GD). For the study, 10 weeks old C57BL/6 mice were randomly divided into normal groups and GD group. The latter group of animals were maintained on a Western Diet two weeks before mating until delivery whereas the animals in the normal group were fed a normal diet. Estrous cycle was assessed by standard vaginal smears from weeks 4 to 8. Blood samples, liver and ovary samples were collected from each group during proestrus. A total of 80 mice were used (20 females divided into 2 groups, and 4 groups of 15 off-springs each). Standard biochemical and immunochemical procedures were used to assess a variety of functional parameters in the animals, including a variety of ER and insulin-signaling related proteins. The results provided in the manuscript indicate that GD-normal group off-springs already presented with insulin resistance and glucose intolerance, and the additional of the Western Diet worsened the condition. Serum estradiol increased and peaked in the normal group at week 7 whereas the levels were lower in the GD mice and the western diet advanced the estradiol peak by 1 week. Cyp19A1 activity was downregulated in GD and resulted in increased follicle atresia, worsened by the western diet. Expression of ER alpha was low in liver of GD off-springs along with decreased IRS-1 and Akt phosphorylation, and the administration of the western diet made these changes worse. Based on these results, the authors concluded that GD exposure contributed to glucose metabolic disorders by affecting the sexual maturity of female off-springs, and that a high-calorie diet exacerbated this dysfunction.
Comments: Overall, the study is properly planned and executed, and the conclusions presented by the authors are supported by the data reported in the manuscript. The English style of the manuscript is appropriate. A couple of minor points, however, need to be clarified.
- The Title needs to be amended to something like "...in the off-springs of Gestational Diabetes dams" as it reads as incomplete as written
- Animal experimental design: It is unclear how many animals were used at start and how many were males versus females. It is unclear whether males animals were also randomly divided between normal diet and western diet. If not, why not? Genetically speaking, the off-springs are the results of two parental units, and it is unclear to which extent the males contributed to the modifications reported in the study.
- On the other hand, if the male mice were also divided between the two groups (normal diet and western diet) why no data relative to male mice were provided in the study. Did the western diet affected their sperm counts, morphology and/or motility? The authors should really clarified which animals were exposed to the two diets (females only? males and females?) to better understand and frame the obtained results.
- It is unclear whether Blood liver and ovary samples (line 93) were obtained from all the animals in that group multiple times during the study or from selected animals at different times during the study, or from selected animals that were euthanized for that purpose. If the samples were taken multiple times from the same animals, how much blood was withdrawn at each time point? From where? Was it replaced with saline? Same question about liver and ovaries. How much tissue was biopsied every time?
- The authors state that the off-springs were divided in groups of 15 for each of the four experimental groups. How many of these 15 off-springs were males and how many were females? Were the provided results obtained only by the females in that group or were they pooled from both males and females? Were there any substantial differences between the two off-spring genders?
Author Response
Comments 1: The Title needs to be amended to something like "...in the off-springs of Gestational Diabetes dams" as it reads as incomplete as written.
Response 1: We sincerely appreciate your comprehensive review of our thesis and the valuable suggestions you provided. Your time and effort in helping us improve its quality and readability are truly commendable. Below are our detailed responses to your review comments, and we have modified the title of the thesis to: High-calorie Diet Exacerbates the Crosstalk Between Gestational Diabetes and Youth-onset Diabetes in Female Offspring through Disrupted Estrogen Signaling.
Comments 2: Animal experimental design: It is unclear how many animals were used at start and how many were males versus females. It is unclear whether males animals were also randomly divided between normal diet and western diet. If not, why not? Genetically speaking, the off-springs are the results of two parental units, and it is unclear to which extent the males contributed to the modifications reported in the study.
Response 2: Thank you so much for your question. Here was our answer to this question. In this study, we mainly focused on the effect of maternal hyperglycemia during pregnancy on adolescent metabolic disorders in female offspring mice, so the influencing factors were limited to maternal GDM. The male mice used for cohabitation in the study were all healthy adult male mice, and all male mice fed a normal diet. According to past experience, the number of embryos in pregnant GDM mice was lower than that in the NC group, and GDM mice were prone to dystocia and stillbirth during childbirth. GDM mice were unstable and often killed newborn offspring mice. In order to obtain the same number of female offspring mice, the number of dams in the normal control group (NC) was 15 and the number of female mice in the GDM group was 20. The GDM model was induced using Western diet during pregnancy, and the OGTT results performed on E16.5 identified that the success rate of the GDM model was approximately 85%.
Comments 3: On the other hand, if the male mice were also divided between the two groups (normal diet and western diet) why no data relative to male mice were provided in the study. Did the western diet affected their sperm counts, morphology and/or motility? The authors should really clarified which animals were exposed to the two diets (females only? males and females?) to better understand and frame the obtained results.
Response 3: Thank you for raising this question. Regarding this question, we thought that in this experiment, all the male mice used for co-housing were healthy adult male mice, and they were provided with a normal diet. The co-housing process was carried out only at night, with a male-to-female ratio of 1 : 2. The male mice were removed the next morning. This part of the content was already explained in the methods section, specifically in lines 115-118.
Comments 4: It is unclear whether Blood liver and ovary samples (line 93) were obtained from all the animals in that group multiple times during the study or from selected animals at different times during the study, or from selected animals that were euthanized for that purpose. If the samples were taken multiple times from the same animals, how much blood was withdrawn at each time point? From where? Was it replaced with saline? Same question about liver and ovaries. How much tissue was biopsied every time?
Response 4: Thank you for pointing out this problem. We indeed didn't mention this point in the paper. We have revised the thesis, elaborated on this issue in detail, and provided corresponding explanations and analyses. The collection of tissue samples from female offspring mice has been described in detail in the method (lines 121-130). Female offspring mice in the NC and GDM groups were divided into groups at 3 weeks of age: NC-NC, GDM-NC, NC-WD and GDM-WD groups. The first two groups were given a normal diet, and the latter two groups were given a WD diet. Samples were collected once a week from week 4 to week 8. The specific sampling process is as follows: 5 female mice in pre-estrus were selected from each group of animals by vaginal smear method, and blood samples, ovaries and liver tissues were collected after euthanasia for subsequent experiments. For the collection of blood samples used for detecting the serum estrogen level: The proestrus stage was determined by the vaginal smear method, and 100 μL of blood was collected from the retro-orbital venous plexus of 8 to 10 mice in each group. For the OGTT and ITT experiments conducted in the 5th and 7th weeks, there were n = 8 in the 5th week and n = 6 in the 7th week.
Comments 5: The authors state that the off-springs were divided in groups of 15 for each of the four experimental groups. How many of these 15 off-springs were males and how many were females? Were the provided results obtained only by the females in that group or were they pooled from both males and females? Were there any substantial differences between the two off-spring genders?
Response 5: Thank you so much for pointing out this issue. We have made the following explanations. According to the 10th edition of Diabetes Map, the incidence of youth onset type 2 diabetes (YOT2D) is higher in girls than in boys, which may be closely related to the difference in sexual hormones. Therefore, female offspring with higher YOT2D incidence were selected in this study. All offspring studies were conducted in female offspring mice rather than male offspring mice. Certainly, male offspring mice may also experience metabolic disorders. In the follow-up, the role of gender in the metabolic disorders of GDM offspring will be studied emphatically, as well as the impact of male parents on the metabolism of their offspring.
Round 2
Reviewer 1 Report
Comments and Suggestions for Authors I would like to thank the authors for providing the revised manuscript and addressing my comments. The improvements made enhance the clarity and overall quality of the work.I appreciate the revisions made so far; however, I must point out that previous report comments 37 and 39 have not been fully addressed:
-
Comment 37: In line 527, the term "previous studies" is mentioned without any supporting citations. To ensure that readers can verify and contextualize the claims made, it is imperative to include appropriate references that substantiate this statement.
-
Comment 39: The manuscript currently lacks a distinct conclusion that encapsulates the preclinical implications of the work and outlines clear future directions. Please add a dedicated concluding section that not only synthesizes the key preclinical findings but also suggests potential avenues for future research. This will help clarify the broader impact of your study and provide guidance for subsequent investigations.
Additionally, in lines 168–169 and lines 178–181, please ensure that the initials of the investigator are clearly indicated. This detail is important for proper attribution and clarity regarding the specific contributions of the investigator in those sections. Including the investigator's initials aligns the manuscript with standard academic reporting practices and enhances overall transparency.
Finally, please note that the ARRIVE checklist has not been updated to reflect the revisions made in the manuscript. Updating this checklist is essential to ensure full compliance with the ARRIVE guidelines and to provide reviewers with an accurate account of the modifications that have been implemented. The ARRIVE checklist should be revised accordingly before further consideration of the manuscript.
Author Response
Comments 37: In line 527, the term "previous studies" is mentioned without any supporting citations. To ensure that readers can verify and contextualize the claims made, it is imperative to include appropriate references that substantiate this statement.
Response 37: Thank you so much for pointing out the deficiencies in the article again. I have added the corresponding references in line 540 of the text.
Wei, W.; Qin, F.; Gao, J.; Chang, J.; Pan, X.; Jiang, X.; Che, L.; Zhuo, Y.; Wu, D.; Xu, S. The effect of maternal consumption of high-fat diet on ovarian development in offspring. Animal Reproduction Science 2023, 255, doi:10.1016/j.anireprosci.2023.107294.
Comments 39: The manuscript currently lacks a distinct conclusion that encapsulates the preclinical implications of the work and outlines clear future directions. Please add a dedicated concluding section that not only synthesizes the key preclinical findings but also suggests potential avenues for future research. This will help clarify the broader impact of your study and provide guidance for subsequent investigations.
Response 39: Thank you for pointing out the shortcomings. We have added a conclusion in the text. It's in lines 579-593.
Conclusion: Embryonic GDM exposure affects the development and maturation of follicles during the sexual maturation, causing the peak of serum estrogen to move forward, but serum estrogen level and ERα signaling is decreased. These are the reasons for the high incidence of glucose metabolism disorders in GDM female-offspring mice. However, judging from the root causes, the intrauterine GDM microenvironment is likely to have a serious impact on the development of the ovaries of female offspring during the embryonic stage, but this effect is obviously manifested during sexual maturity. Female offspring of GDM carry the mark of high metabolic stress from the embryo, and its adverse effects on ovarian development and maturation exacerbate the occurrence of youth-onset type 2 diabetes. In future research, we will focus on the impact of GDM's hypermetabolic microenvironment on the development of the ovary of female offspring during the embryonic period, and how to adjust the diet of pregnant women with GDM to minimize the adverse outcomes of GDM on the ovarian development of female offspring mice.
Comments: Additionally, in lines 168–169 and lines 178–181, please ensure that the initials of the investigator are clearly indicated. This detail is important for proper attribution and clarity regarding the specific contributions of the investigator in those sections. Including the investigator's initials aligns the manuscript with standard academic reporting practices and enhances overall transparency.
Response: This issue is indeed something we haven't noticed. We have made modifications in lines 169-171 and 181-185 of the text. Modify as follows:
The above procedure was performed by researchers Xiangju Cao and Yuan Wang, who were blinded to group assignment, with data subsequently analyzed by an independent researcher, Xinyu Jia.
First, researchers Xiangju Cao and Yuan Wang classified follicles into morphological categories while blinded to group assignment and counted the follicles at each stage. The counting results were then independently subjected to statistical analysis by a separate researcher, Xinyu Jia.
Finally, thank you for your reminder. I have uploaded the revised ARRIVE checklist.
Reviewer 2 Report
Comments and Suggestions for Authors
In this study, the authors investigated the role of estrogen signaling in youth-onset diabetes in the off-springs of dams with gestational diabetes (GD). For the study, 10 weeks old C57BL/6 mice were randomly divided into normal groups and GD group. The latter group of animals were maintained on a Western Diet two weeks before mating until delivery whereas the animals in the normal group were fed a normal diet. Estrous cycle was assessed by standard vaginal smears from weeks 4 to 8. Blood samples, liver and ovary samples were collected from each group during proestrus. A total of 80 mice were used (20 females divided into 2 groups, and 4 groups of 15 off-springs each). Standard biochemical and immunochemical procedures were used to assess a variety of functional parameters in the animals, including a variety of ER and insulin-signaling related proteins. The results provided in the manuscript indicate that GD-normal group off-springs already presented with insulin resistance and glucose intolerance, and the additional of the Western Diet worsened the condition. Serum estradiol increased and peaked in the normal group at week 7 whereas the levels were lower in the GD mice and the western diet advanced the estradiol peak by 1 week. Cyp19A1 activity was downregulated in GD and resulted in increased follicle atresia, worsened by the western diet. Expression of ER alpha was low in liver of GD off-springs along with decreased IRS-1 and Akt phosphorylation, and the administration of the western diet made these changes worse. Based on these results, the authors concluded that GD exposure contributed to glucose metabolic disorders by affecting the sexual maturity of female off-springs, and that a high-calorie diet exacerbated this dysfunction.
Comments: In this revision of their study, the authors have addressed satisfactorily the comments and the clarifications requested during the previous reviewing cycle. No additional comments or criticisms were raised at this time.
Author Response
Comments: In this revision of their study, the authors have addressed satisfactorily the comments and the clarifications requested during the previous reviewing cycle. No additional comments or criticisms were raised at this time.
Response: Thank you so much for your reply.